# PhysCodeBench: Benchmarking Physics-Aware Symbolic Simulation of 3D Scenes via Self-Corrective Multi-Agent Refinement

## Abstract

Physics-aware symbolic simulation of 3D scenes is critical for robotics, embodied AI, and scientific computing, requiring models to understand natural language descriptions of physical phenomena and translate them into executable simulation environments. While large language models (LLMs) excel at general code generation, they struggle with the semantic gap between physical descriptions and simulation implementation. We introduce PhysCodeBench, the first comprehensive benchmark for evaluating physics-aware symbolic simulation, comprising 700 manually-crafted diverse samples across mechanics, fluid dynamics, and soft-body physics with expert annotations. Our evaluation framework measures both code executability and physical accuracy through automated and visual assessment. Building on this, we propose a Self-Corrective Multi-Agent Refinement Framework (SMRF) with three specialized agents (simulation generator, error corrector, and simulation refiner) that collaborate iteratively with domain-specific validation to produce physically accurate simulations. SMRF achieves 67.7 points overall performance compared to 36.3 points for the best baseline among evaluated SOTA models, representing a 31.4-point improvement. Our analysis demonstrates that error correction is critical for accurate physics-aware symbolic simulation and that specialized multi-agent approaches significantly outperform single-agent methods across the tested physical domains.

## 1 Introduction

Physics-aware symbolic simulation of 3D scenes has become a cornerstone of modern robotics, embodied AI, and scientific computing. From simulating robot dynamics for motion planning to modeling fluid flows for climate prediction, the ability to accurately encode physical laws in executable code is increasingly critical. However, creating such code remains a formidable challenge that requires both deep programming expertise and sophisticated understanding of physical laws.

By symbolic simulation, we refer to code generation that creates symbolic representations of physical 3D scenes through executable simulation code, enabling computational modeling of complex physical phenomena. The complexity of physics-aware symbolic simulation stems from multiple interacting challenges that single-agent approaches struggle to address simultaneously. First, models must translate natural language descriptions of physical phenomena into syntactically correct code that executes without runtime errors. Second, they must ensure that the generated code accurately implements the intended physical laws and parameters. Third, they must handle simulation boundary conditions and numerical stability constraints that are crucial for realistic behavior. These challenges are fundamentally different in nature (syntactic correctness, physical accuracy, and numerical stability), making it difficult for single models to optimize across all dimensions effectively.

Consider seemingly simple tasks like creating a cube sliding down an incline, simulating a ball bouncing on a trampoline, or generating raindrops falling on a surface. These scenarios require precise implementation of gravity, elasticity, surface tension, and collision dynamics. As illustrated in Figure 1, vanilla LLMs struggle with these tasks, producing failed simulations, implementation bugs, inaccurate results, and incorrect physics parameters. Our preliminary evaluation reveals that state-of-the-art models can only produce executable code at relatively low success rates. Even

Figure 1: **PhysCodeBench and SMRF enable accurate physics-aware symbolic simulation of 3D scenes.** Our Self-Corrective Multi-Agent Refinement Framework (SMRF) achieves 67.7 points overall performance, significantly outperforming state-of-the-art models (best baseline: 36.3 points). The framework correctly implements complex physics like conservation laws, surface tension, and collision dynamics, resulting in a remarkable 31.4-point improvement over existing approaches.

when the code executes successfully, GPT-4o (Hurst et al., 2024) and Claude-3.5-Sonnet (Anthropic, 2024) frequently generate simulations where object interactions deviate from the instruction descriptions, producing inappropriate physics parameters, inaccurate boundary conditions, and incorrect object positioning that result in behaviors inconsistent with the intended physical scenarios. These models demonstrate a fundamental challenge: achieving basic code execution is only the first hurdle, as the generated simulations often fail to match the physical phenomena described in the natural language instructions.

This failure points to a fundamental limitation: existing code generation models excel at syntactic correctness but struggle with the semantic requirements of physical accuracy and the practical constraints of numerical implementation. While general code generation has been extensively studied (Chen et al., 2021; Roziere et al., 2023), the specialized domain of physics-aware symbolic simulation presents unique challenges that remain largely unaddressed. Current benchmarks focus on general programming tasks and lack the domain-specific evaluation criteria necessary to assess physics-aware symbolic simulation quality.

The multi-faceted nature of this challenge requiring expertise in scene composition, physics understanding, and spatial reasoning motivates a decomposition approach where specialized agents can focus on their respective strengths. However, the lack of comprehensive benchmarks for physics-aware symbolic simulation has hindered progress in this critical area.

We address this gap with three key contributions:

- **PhysCodeBench**: The first comprehensive benchmark specifically designed for physics-aware symbolic simulation of 3D scenes, featuring manually-crafted examples across solid mechanics, fluid dynamics, and soft-body physics domains with expert annotations.

- **Self-Corrective Multi-Agent Refinement Framework (SMRF)**: A novel multi-agent system that decomposes physics-aware symbolic simulation of 3D scenes into specialized components: a simulation generator for initial implementation, an error corrector for diagnosing and fixing errors, and a simulation refiner for optimization and preference alignment. This achieves a remarkable 31.4-point improvement over state-of-the-art models.

- **Comprehensive Evaluation and Analysis**: Through extensive experiments and user studies, we demonstrate that SMRF not only generates more physically accurate simulations but also produces code that domain experts rate significantly higher in quality and practical utility.

Our experiments demonstrate that decomposing physics-aware symbolic simulation into specialized agents (each focusing on code generation, error correction, and code refinement) enables more accurate implementation of complex physical phenomena. This work establishes a new foundation for AI systems that can bridge the gap between physical understanding and executable implementation, a capability that will be crucial as autonomous systems become more prevalent in physical environments.

## 2 RELATED WORK

Our work bridges physical simulation understanding, code generation, and multi-agent systems. We review key advances and identify gaps that PhysCodeBench and SMRF address.

### 2.1 PHYSICS SIMULATION AND SYMBOLIC CODE GENERATION

Physical reasoning has been studied through video prediction tasks like Physion (Bear et al., 2021) and CLEVRER (Yi et al., 2019), which focus on understanding phenomena rather than code implementation. Domain-specific benchmarks include PDEBench (Takamoto et al., 2022) for differential equations and PHYBench (Qiu et al., 2025) for physics problem solving, but none address physics-aware symbolic simulation.

Recent work explores LLM-based code generation for robotics. Mind's Eye (Liu et al., 2022) uses physics simulation to enhance language model reasoning through bilevel optimization with Mu-JoCo (Todorov et al., 2012). Other robotics applications leverage LLMs to generate code for task execution: Code as Policies (Arenas et al., 2024) writes hierarchical robot control policies, Vox-Poser (Huang et al., 2023) composes 3D value maps for motion planning, and SimGen (Zhou et al., 2024) generates realistic driving scene images. However, these focus on high-level task planning or scene generation within constrained domains. Our work addresses generating fundamental physics simulation code that symbolically represents physical laws accurately across broader 3D scene domains.

Code generation has advanced with models like Codex (Chen et al., 2021), CodeLlama (Roziere et al., 2023), DeepSeek-R1 (Guo et al., 2025), and Qwen-2.5 (Yang et al., 2024). These are evaluated on general programming benchmarks (HumanEval (Chen et al., 2021), MBPP (Austin et al., 2021)) but lack domain-specific evaluation for physics simulation, where syntactic correctness must couple with physical accuracy. Fine-tuning approaches like WizardCoder (Luo et al., 2023) show specialization improves domain performance, motivating our multi-agent approach.

### 2.2 MULTI-AGENT SYSTEMS AND PREFERENCE ALIGNMENT

Multi-agent frameworks like AutoGen (Wu et al., 2023) and ChatDev (Qian et al., 2023) demonstrate how specialized agents collaborate on complex tasks. In code generation, approaches like Self-Debugging (Chen et al., 2023) and CodeChain (Le et al., 2023) show iterative refinement improves quality, but lack physics-specific knowledge. Multi-agent approaches have shown promise in specialized domains like personalized healthcare applications (Liu, 2025), demonstrating that complex tasks benefit from decomposed agent architectures.

Preference alignment techniques (RLHF (Ouyang et al., 2022), DPO (Rafailov et al., 2023)) have improved LLM quality generally, but remain limited in specialized domains like physics simulation. Our SMRF extends these ideas by designing physics-specific agent roles and applying DPO with expert preferences for physics-aware symbolic simulation.

## 3 PHYSCODEBENCH DATASET

### 3.1 OVERVIEW

We introduce PhysCodeBench, a comprehensive benchmark for evaluating AI models' ability to generate physically accurate symbolic simulation code for 3D scenes. Our dataset comprises 700 examples spanning rigid-body physics, soft-body physics, fluid dynamics, and mechanics (Figure 2). Each example includes detailed metadata: difficulty levels, physical laws, and human preference ratings. The dataset is split into 600 training and 100 testing examples with balanced domain coverage.

Figure 2 shows representative examples from each domain: cylinder rolling (rigid-body), trampoline with bouncing ball (soft-body), and raindrop interactions (fluid dynamics). This diversity ensures comprehensive coverage of fundamental physical phenomena. Detailed dataset statistics are provided in Appendix A.1.1.

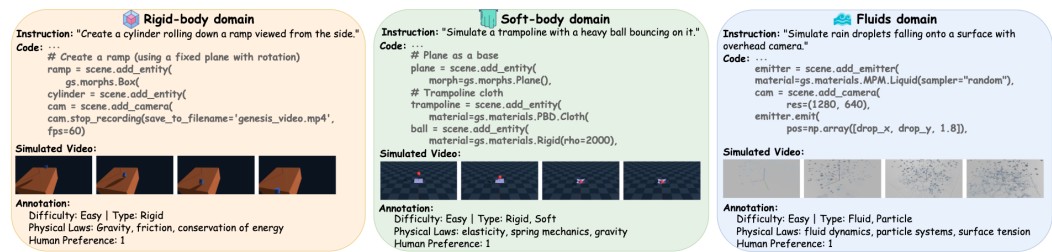

Figure 2: Examples from the PhysCodeBench dataset spanning different physical domains. Each example includes the instruction prompt, key code snippets, simulated video frames, and comprehensive annotation metadata.

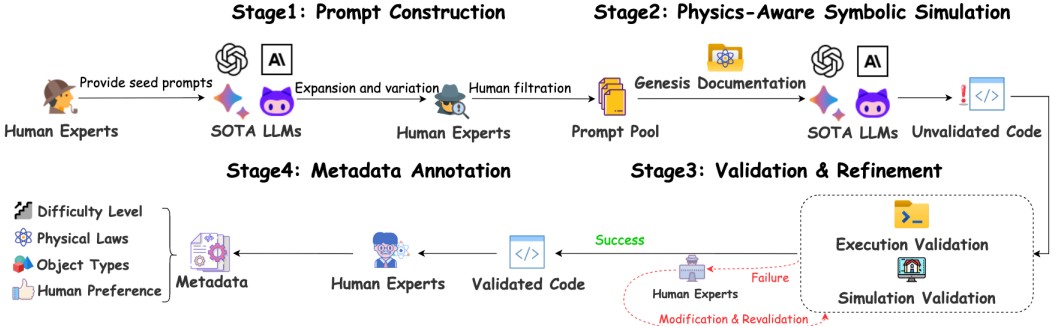

Figure 3: PhysCodeBench dataset curation pipeline. Our four-stage process involves prompt construction, physics-aware symbolic simulation, validation and refinement, and metadata annotation.

## 3.2 DATASET CURATION PROCESS

Our dataset construction follows a rigorous four-stage pipeline (Figure 3) ensuring both physical accuracy and programmatic validity.

### 3.2.1 PROMPT CONSTRUCTION

The initial stage involves creating natural language prompts that describe physical scenarios across different categories, difficulty levels, physical laws, and simulation phenomena. Human experts first draft seed prompts covering the desired range of scenarios. These seed prompts are then provided to multiple state-of-the-art proprietary models for expansion and variation, enabling the generation of a large number of instructions from a limited set of seed prompts. Human experts subsequently filter and select suitable simulation instructions from these machine-generated candidates to form the Prompt Pool. Through this process, we generated 1,000 instruction candidates from 50 carefully crafted seed prompts.

### 3.2.2 PHYSICS-AWARE SYMBOLIC SIMULATION

For each validated prompt from the Prompt Pool, we generate corresponding simulation code using the proprietary models. To ensure diversity and reduce potential biases, we employed multiple large language models including Github Copilot (Github, 2025), Claude-3.5-Sonnet (Anthropic, 2024), Gemini-2.0-Pro (Gemini, 2024), and GPT-4o (Hurst et al., 2024) to generate the code samples. The input to these LLMs includes: the validated prompt, Genesis code library documentation, and specific formatting requirements. While the generated code from different sources exhibited minor stylistic variations, all implementations shared the same fundamental structure and functionality as they all needed to execute within the Genesis environment to produce simulation videos. This approach enables us to create a more balanced dataset that is not biased toward any specific model's coding style or implementation preferences. The detailed prompt template used for physics-aware symbolic simulation is provided in Appendix A.1.4.

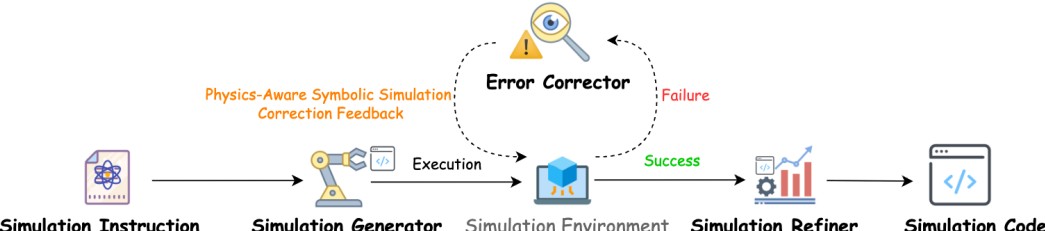

Figure 4: The Self-Corrective Multi-Agent Refinement Framework (SMRF) features three specialized agents: Simulation Generator, Error Corrector, and Simulation Refiner. The framework creates a feedback loop where each agent performs a specific role in generating, correcting and refining physics-aware symbolic simulation code.

### 3.2.3 CODE VALIDATION AND REFINEMENT

The generated code (unvalidated code) undergoes a two-stage validation process: execution validation and physical simulation validation. Execution validation tests whether the code executes without errors in the Genesis environment. Physical simulation validation examines whether the executed code produces appropriate simulation visualization files that match the intended physical scenario. Code that passes both validations becomes validated code and is paired with its corresponding prompt to form a dataset entry.

When validation fails, human experts manually modify the code and repeat the validation process, with up to three modification attempts allowed. Code that fails to pass validation after three attempts is discarded. Through this rigorous process, only 30% of the generated code (300 out of 1,000 prompts) successfully passes validation to become validated code.

To prepare preference data for training, we generate two different validated code implementations for the same prompt. Human experts subsequently evaluate both implementations based on the prompt requirements and simulation results, selecting the preferred code for preference learning.

### 3.2.4 METADATA ANNOTATION

After validating the prompt-code pairs, we enrich each example with comprehensive metadata to facilitate targeted evaluation and analysis. As shown in Figure 2, this metadata includes difficulty level, physical laws (the key physics concepts demonstrated), object types (categories of objects involved), and human preference ratings (quality scores assigned by human evaluators to the generated simulations). The detailed annotation guidelines, can be found in Appendix A.1.5.

## 4 METHODOLOGY

To enhance physics-aware symbolic simulation of 3D scenes capabilities, we develop an innovative multi-agent approach called the Self-Corrective Multi-Agent Refinement Framework (SMRF).

### 4.1 SELF-CORRECTIVE MULTI-AGENT REFINEMENT FRAMEWORK (SMRF)

Our novel SMRF introduces a multi-agent collaborative approach where specialized agents work together to generate, correct, and improve physics-aware symbolic simulation. As shown in Figure 4, the framework consists of three key components:

The **Simulation Generator (SG)** is trained on PhysCodeBench's training set using supervised fine-tuning and produces initial code based on the instruction prompt. The generated code is executed in a Python environment within the Genesis physics engine to identify runtime errors or execution failures. When validation fails, the **Error Corrector (EC)** diagnoses specific errors and proposes corrections based on its training on tetrads of (instruction, incorrect code, error description, correct code). The EC attempts to correct errors up to 3 times; if error persist after these attempts, the framework cannot process the prompt and returns a failure. When execution succeeds, the **Simula-**

**tion Refiner (SR)** further optimizes the code by implementing corrections while maintaining code structure and optimizing for human preferences through preference-based training.

The theoretical foundation for our multi-agent approach stems from three key insights about physics-aware symbolic simulation. **First**, physical simulation code requires integrating programming expertise with physics understanding—domains that single-agent approaches struggle to simultaneously optimize, often producing executable but physically inaccurate simulation code. **Second**, detecting and correcting physical errors requires specialized knowledge beyond general code generation. Single fine-tuned agents can improve generation success rates but fail to identify deeper issues like parameter errors or conservation law violations that only emerge during execution. **Third**, our approach applies the cognitive science principle of specialized modules communicating through structured protocols (Carruthers, 2006; Robbins, 2009). This allows each agent to develop domain-specific expertise while collaborating effectively, creating an iterative self-correcting system that addresses challenges beyond any individual component's capabilities.

## 4.2 SUPERVISED FINE-TUNING (SFT) AND PREFERENCE ALIGNMENT

We perform specialized supervised fine-tuning for each agent using our training set of 600 examples, with data formats tailored to each agent's role:

The **Simulation Generator (SG)** is trained on instruction-code pairs, learning to map natural language descriptions of physical scenarios to their corresponding implementation code. The training objective follows standard supervised learning (Chen et al., 2021):

$$\mathcal{L}_{\text{SFT}} = - \sum_{i=1}^{N} \log p_\theta(y_i|x_i) \tag{1}$$

where $x_i$ represents the instruction prompt, $y_i$ is the reference implementation, and $p_\theta$ is the model's conditional probability distribution. This training approach helps models learn the mapping from natural language descriptions of physical scenarios to correct code implementations.

The **Error Corrector (EC)** is trained on tetrad of (instruction, incorrect code, error description, modified correct code), learning to diagnose specific physics-related errors and suggest appropriate corrections. This specialized training helps the EC identify subtle physical violations that might not cause execution errors but would result in physically unrealistic simulations.

The **Simulation Refiner (SR)** is first trained using SFT on instruction-code pairs, then further refined using direct preference optimization (DPO) (Rafailov et al., 2023) to align with human preferences for code quality and style. The DPO objective is:

$$\mathcal{L}_{\text{DPO}} = -\mathbb{E}_{(x,y_w,y_l)\sim\mathcal{D}} \left[ \log \sigma \left( \beta \log \frac{p_\theta(y_w|x)}{p_{\text{ref}}(y_w|x)} - \beta \log \frac{p_\theta(y_l|x)}{p_{\text{ref}}(y_l|x)} \right) \right] \tag{2}$$

where $y_w$ and $y_l$ are the preferred and less preferred implementations respectively, $p_{\text{ref}}$ is the reference model's probability, and $\beta$ is a hyperparameter controlling the strength of the preference. Detailed information on the training data preparation, hyperparameters, and training procedures for each agent can be found in Appendix A.2.

## 4.3 EVALUATION METRICS

To comprehensively assess the quality of generated physical simulations, we develop PhysCodeEval, a dual-faceted evaluation framework with maximum score of 100 points: Total $= S_{\text{code}} + S_{\text{visual}}$, as detailed in Appendix A.4.

The code quality score $S_{\text{code}}$ (50 points) evaluates whether the generated code executes successfully ($S_{\text{exec}}$, 25 points) and produces the expected simulation files ($S_{\text{file}}$, 25 points), with each component awarding full points for success and zero for failure.

The simulation fidelity score $S_{\text{visual}}$ (50 points) assesses the generated simulation quality through two metrics: $S_{\text{clip}}$ measures semantic alignment between simulation video and instruction using ClipScore (Hessel et al., 2021), while $S_{\text{motion}}$ evaluates physical realism via motion smoothness assessment (Huang et al., 2024) to detect anomalies such as jitter, unrealistic accelerations, or object intersections. Each component contributes up to 25 points.

## 5 EXPERIMENTS

### 5.1 EXPERIMENTAL SETUP

**Models and Baselines.** We compare our approach against several strong baselines. For proprietary models, we evaluate GPT-4o (Hurst et al., 2024), Claude-3.5-Sonnet (Anthropic, 2024), and Gemini-2.0-Pro (Gemini, 2024), representing current state-of-the-art general-purpose LLMs. From the open-source community, we test DeepSeek-R1 (Guo et al., 2025), DeepSeek-R1-Distill-Qwen-32B (DeepSeek, 2025), Qwen-2.5-32B (Yang et al., 2024), and QwQ-32B (Team, 2025) all high-performing code generation models. We also create fine-tuned single-agent variants of these base models using SFT and DPO (Rafailov et al., 2023), which operate without the multi-agent framework. Our SMRF implementation includes three variants: SMRF (base), which uses unspecialized base models; SMRF + SFT, which incorporates supervised fine-tuning; and SMRF + SFT + DPO, our complete framework that adds preference alignment to Simulation Refiner Agent.

**Training and Inference.** For SMRF, all agents are initialized from DeepSeek-R1-Distill-Qwen-32B and specialized through role-specific training. We use a learning rate of 1e-5, batch size of 2, and train for 5 epochs for SFT and 3 epochs for DPO. Training was conducted on 2 NVIDIA A100 GPUs with 80GB memory, requiring approximately 20 GPU-hours.

During inference, each model is provided with comprehensive Genesis physics engine documentation ($\sim$100K tokens), including API references and examples. Each model generates responses with a maximum length of 4096 tokens and temperature of 0.1. We perform 5 inference passes per test prompt and report the average score. Complete training and inference details are provided in Appendix A.2.

**Evaluation.** We evaluate all approaches using our comprehensive evaluation framework introduced in Section 4.3 on PhysCodeBench testing data. The Code-based evaluation (50 points) consists of Code Executability (CE, 25 points) and File Generation (FG, 25 points). The Visual-based evaluation (50 points) includes Clip Similarity (CS, 25 points) and Motion Smoothness (MS, 25 points). The Combined Score (Total) is the sum of these four components, with a maximum of 100 points.

### 5.2 QUANTITATIVE ANALYSIS

Table 1 presents the performance of all approaches on the PhysCodeBench test set. Several key findings emerge from these results. First, our SMRF framework consistently outperforms single-agent approaches, even when comparing SMRF (base) to fine-tuned single agents. This demonstrates the value of task decomposition for physics-aware symbolic simulation. Second, SMRF + SFT + DPO achieves the highest overall performance across all metrics, reaching 67.7 points (out of 100) total score and surpassing the best proprietary model (Claude-3.5-Sonnet at 36.3 points) by a substantial margin of 31.4 points. Third, while SFT alone provides substantial improvements (increasing performance from 34.1 to 55.7 points), adding DPO further enhances performance by an additional 12.0 points (from 55.7 to 67.7 points) by aligning the Simulation Refiner's behavior with expert preferences for physics-aware symbolic simulation code quality. Single-agent iterative refinement (3 attempts) improves to 41.3 points but remains 26.4 points below SMRF (details in Appendix A.5.6).

### 5.3 ABLATION STUDIES

To understand the contribution of each component, we conducted ablation studies by removing key elements of the SMRF framework:

Table 2 reveals the importance of each SMRF component. Removing the Simulation Refiner (SR) causes an 8.8 point drop, highlighting its role in optimizing code and preference alignment. The Error Corrector (EC) contributes 12.0 points by implementing corrections while maintaining code

Table 1: Performance comparison on PhysCodeBench test set. Code-based (0-50 points): execution & file generation; Visual-based (0-50 points): simulation fidelity. Total score ranges from 0-100 points. DRDQ-32B = DeepSeek-R1-Distill-Qwen-32B.

| Model | Code-based | Visual-based | Total |
|---|---|---|---|
| *Vanilla Proprietary Models (Zero-shot)* | | | |
| GPT-4o | 16.0 | 18.3 | 34.3 |
| Claude-3.5-Sonnet | 17.2 | 19.1 | 36.3 |
| Gemini-2.0-Pro | 15.0 | 17.0 | 32.0 |
| *Vanilla Open-source Models (Zero-shot)* | | | |
| DeepSeek-R1 | 14.0 | 15.8 | 29.8 |
| DRDQ-32B (vanilla) | 12.2 | 15.8 | 28.0 |
| Qwen-2.5-32B | 0.7 | 1.1 | 1.8 |
| QwQ-32B | 6.8 | 9.0 | 15.8 |
| *Single-Agent Fine-tuned* | | | |
| DRDQ-32B + SFT | 17.5 | 18.4 | 35.9 |
| DRDQ-32B + SFT + DPO | 18.7 | 19.2 | 37.9 |
| DRDQ-32B + SFT + DPO (w/ iter. refinement) | 20.1 | 21.2 | 41.3 |
| *Our Multi-Agent Framework (SMRF)* | | | |
| Base (vanilla agents) | 16.4 | 17.7 | 34.1 |
| + SFT | 27.3 | 28.4 | 55.7 |
| + SFT + DPO | **33.5** | **34.2** | **67.7** |

Table 2: Ablation study of SMRF components. Mean performance across 5 runs with different random seeds.

| Configuration | Code-based | Visual-based | Total |
|---|---|---|---|
| Full SMRF + SFT + DPO | 33.5 | 34.2 | 67.7 |
| - Without SR | 29.1 | 29.8 | 58.9 |
| - Without EC | 27.3 | 28.4 | 55.7 |
| - SFT only (SR without DPO) | 30.8 | 31.2 | 62.0 |

coherence. DPO alignment provides a 5.7 points boost over SFT-only training, confirming the value of preference optimization. These results demonstrate how specialized components collectively address different aspects of physics-aware symbolic simulation.

## 5.4 QUALITATIVE ANALYSIS

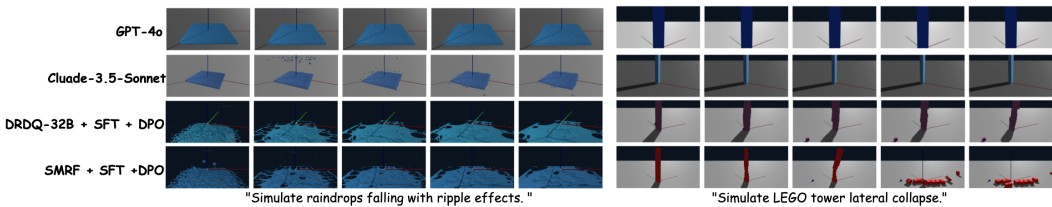

Figure 5: Qualitative comparison of physical simulations generated by different approaches for two test prompts. Left: "Simulate raindrops falling with ripple effects." Right: "Simulate LEGO tower lateral collapse." For each prompt, we show frames from simulations by GPT-4o, Claude-3.5-Sonnet, DRDQ-32B + SFT + DPO, and our SMRF + SFT + DPO approach (left to right).

Our qualitative analysis reveals significant differences in physical realism. For fluid dynamics (Figure 5, left), GPT-4o and Claude-3.5-Sonnet produce simplified water surfaces with limited ripple propagation, while SMRF generates realistic surface tension, splash formation, and wave propagation. For rigid body dynamics (right), only SMRF accurately captures the complex collapse dynamics with realistic tumbling and contact physics. Baseline models produce oversimplified collapses or fail to maintain physical constraints, highlighting how SMRF's multi-agent verification leads to substantially more accurate simulations. Additional qualitative results are provided in Appendix A.5.5.

## 5.5 USER STUDY AND METRIC VALIDATION

We conducted a user study with 10 participants (6 students, 4 professional developers) with physics simulation experience. From our test set, we identified prompts where all three models (Claude-3.5-Sonnet, DRDQ-32B + SFT + DPO, and SMRF + SFT + DPO) successfully generated executable code that produced visualization files. To ensure sufficient evaluation data, we supplemented the test set with additional manually constructed instructions until we collected 50 prompts where all models produced valid implementations. We randomly selected 40 of these prompts for evaluation. For each prompt, participants rated the three implementations on a 1-5 scale across three dimensions. Physical accuracy was evaluated based on how well the simulation visualization matched the instruction and the smoothness of the resulting animation. Code readability assessed the overall quality and clarity of the generated code. Overall usefulness provided a holistic evaluation considering both the simulation results and code quality.

Table 3: Results from our user study comparing different approaches based on human evaluations.

| Model | Physical Accuracy | Code Readability | Overall Usefulness |
|---|---|---|---|
| Claude-3.5-Sonnet | 3.2 | 3.7 | 3.4 |
| DRDQ-32B + SFT + DPO | 3.4 | 3.5 | 3.6 |
| SMRF + SFT + DPO | **4.5** | **4.2** | **4.6** |

SMRF significantly outperformed other approaches across all metrics (Table 3). To validate our automatic metrics, we computed Spearman correlations with human judgments: ClipScore (Hessel et al., 2021) correlated strongly with physical accuracy ratings ($\rho = 0.82$), and code execution with usefulness ratings ($\rho = 0.76$). These results confirm our evaluation framework aligns with human perceptions of quality.

## 5.6 ADDITIONAL EXPERIMENTAL ANALYSIS

Our comprehensive evaluation reveals several important insights about SMRF's performance characteristics. The advantage of SMRF over baselines increases with task complexity, suggesting its particular value for challenging physical problems (see Appendix A.5.1). Performance varies significantly across physical domains, with all approaches performing best on solid mechanics tasks and finding fluid dynamics most challenging, yet SMRF maintains consistent advantages across all domains (see Appendix A.5.2). The framework successfully identifies and corrects various types of physical errors, including conservation law violations, parameter miscalibrations, and boundary condition errors, with high success rates (see Appendix A.5.3). Through detailed case studies spanning rigid-body physics, fluid simulations, and soft-body deformations, we demonstrate how our multi-agent approach systematically improves physics-aware symbolic simulation across diverse scenarios (see Appendix A.5.4).

## 6 CONCLUSION

We introduce PhysCodeBench, the first comprehensive benchmark for physics-aware symbolic simulation of 3D scenes, comprising 700 examples across mechanics, fluid dynamics, and soft-body physics domains. Our Self-Corrective Multi-Agent Refinement Framework (SMRF) decomposes the complex task into specialized components for code generation, error correction, and refinement, achieving 67.7 points overall performance and substantially outperforming state-of-the-art models by 31.4 points. Through extensive evaluation and user studies, we demonstrate that specialized multi-agent approach significantly improve physics simulations generation across diverse domain.

While our work focuses on symbolic simulation, this provides essential building blocks for more complex physics programming. Future work will target lower-level physics implementation through PhysCodeBench v2. To facilitate further research, we will release the PhysCodeBench dataset, SMRF implementation, and evaluation framework. This work demonstrates the potential of multi-agent approaches for domain-specific programming tasks.

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

# A  APPENDIX

## A.1  PHYSCODEBENCH DATASET DETAILS

### A.1.1  DATASET STATISTICS

PhysCodeBench consists of 700 examples spanning multiple physical domains. Table 4 shows the distribution of examples across different physical domains and difficulty levels in our dataset. Note that some examples may belong to multiple domains (e.g., a simulation may combine rigid-body physics with fluid dynamics).

Table 4: Distribution of examples across domains and difficulty levels in PhysCodeBench.

| Domain | Easy | Hard | Total |
|---|---|---|---|
| Rigid-body Physics | 144 | 96 | 240 |
| Soft-body Physics | 108 | 72 | 180 |
| Fluid Dynamics | 84 | 76 | 160 |
| Mechanics | 72 | 48 | 120 |
| **Total Primary Classifications** | 408 | 292 | 700 |
| **Multi-domain Examples** | 82 | 138 | 220 |

We further analyze the physical laws involved across the dataset, as shown in Table 5. The most prevalent laws include collisions, gravity, elasticity, friction, and fluid dynamics. Each example typically involves 2-4 physical laws that must be correctly implemented for accurate simulation. Similar to domain classifications, many examples incorporate multiple physical principles simultaneously.

Table 5: Distribution of physical laws across the PhysCodeBench dataset. The table shows the occurrence of each physical principle across domains, with many examples implementing multiple laws.

| Physical Law | Rigid-body | Soft-body | Fluid | Mechanics |
|---|---|---|---|---|
| Collisions | 228 | 95 | 38 | 78 |
| Gravity | 182 | 156 | 142 | 94 |
| Elasticity | 112 | 175 | 24 | 58 |
| Friction | 186 | 92 | 45 | 88 |
| Fluid Dynamics | 16 | 32 | 160 | 12 |
| Other | 34 | 26 | 42 | 24 |

The "Other" category includes specialized physical principles such as surface tension, buoyancy, magnetic interactions, thermal effects, and material phase transitions that appear less frequently in the dataset. As shown in the table, many examples implement multiple physical laws simultaneously—for instance, nearly all rigid-body simulations involve both collision mechanics and gravitational forces.

### A.1.2  EXPERT TEAM COMPOSITION

PhysCodeBench dataset construction involved a multidisciplinary team of 5 domain experts, ensuring comprehensive coverage of physics domains and technical validation:

**Physics Experts (3 members):**

- **Expert 1**: Ph.D. in Mechanical Engineering with specialization in classical mechanics and rigid-body dynamics. Contributed to rigid-body physics examples and validation of collision mechanics, conservation laws, and constraint satisfaction.

- **Expert 2**: Ph.D. in Fluid Dynamics with expertise in computational fluid dynamics (CFD) and Navier-Stokes equations. Responsible for fluid simulation examples, including surface tension, viscosity, and multi-phase flow scenarios.

- **Expert 3**: M.S. in Applied Physics with focus on soft-body mechanics and material science. Contributed to soft-body physics examples involving elasticity, plasticity, and deformation dynamics.

**Computer Graphics Experts (2 members):**

- **Expert 4**: Ph.D. in Computer Science with specialization in physics-based simulation and rendering. Experience includes 2+ years developing physics engines for robotics and animation. Responsible for API correctness validation and simulation quality assessment.

- **Expert 5**: M.S. in Computer Graphics with expertise in real-time simulation systems. Contributed to code quality evaluation, performance optimization assessment, and visual realism validation.

**Collaborative Workflow:** Each example underwent review by at least 2 experts from different backgrounds. Physics experts validated physical accuracy and parameter reasonableness, while computer graphics experts ensured correct API usage and code quality. Disagreements were resolved through group discussion until consensus was reached. This interdisciplinary validation process ensured that generated simulations were both physically accurate and technically correct.

The team's diverse expertise enabled comprehensive evaluation across multiple dimensions: physical law adherence (mechanics, thermodynamics, fluid dynamics), numerical stability, API correctness, code readability, and visual plausibility. This rigorous validation distinguishes PhysCodeBench from automatically generated datasets and ensures high-quality ground truth for evaluating physics-aware code generation capabilities.

### A.1.3    DATA COLLECTION PROCESS

Our data collection process involved a multi-stage pipeline to ensure high-quality examples. In the prompt construction phase, we began with 50 seed prompts created by domain experts. These were expanded to 1,000 candidate prompts using LLMs, from which 700 were selected after human filtering for clarity and diversity.

During code generation and validation, we tracked the success rates at different stages as shown in Table 6. The initial LLM generation success rate (64.2%) reflects the percentage of code generated by state-of-the-art LLMs that passed the execution validation step without runtime errors. However, even executable code often required further validation to ensure physical accuracy. The code that passed execution validation then underwent physical simulation validation, where we examined whether the simulation correctly represented the intended physical phenomena. Both validation steps needed to be passed for code to be considered validated.

When validation failed at either step, human experts intervened to modify the code, increasing the success rate to 82.7%. For remaining issues, we attempted up to three rounds of regeneration, bringing the success rate to 91.6%. The final inclusion rate of 83.3% reflects examples that not only executed correctly but also produced physically accurate simulations, as determined by domain experts. Note that this rate is calculated across all generation rounds: in the initial generation, 30% of code (300/1,000) passed validation; through iterative regeneration with expert correction, we ultimately achieved 700 high-quality examples with an overall 83.3% success rate.

For prompts that required regeneration, we analyzed the types of issues encountered, as detailed in Table 7. API usage errors constituted the largest category (42.8%), reflecting the complexity of implementing the Genesis physics engine correctly. Physical parameter misconfiguration (28.6%) was the second most common issue, highlighting the challenge of translating physical principles into appropriate numerical values.

We observed that more complex physical domains (fluid dynamics and mechanics) required more regeneration attempts on average (2.4 and 2.1 respectively) compared to rigid-body physics (1.3) and soft-body physics (1.8). Through this rigorous process, only approximate 30% of the generated code

Table 6: Success rates at different stages of the data collection pipeline.

| Stage | Success Rate (%) | Common Failure Reasons |
|---|---|---|
| Initial LLM Generation | 64.2 | API misuse, missing parameters |
| After Human Intervention | 82.7 | Complex physics constraints |
| After Regeneration Attempts | 91.6 | Fundamental inconsistencies |
| Final Inclusion | 83.3 | Physics inaccuracies in simulation |

Table 7: Types of issues requiring code regeneration.

| Issue Type | Percentage (%) |
|---|---|
| Syntax Errors | 15.3 |
| API Usage Errors | 42.8 |
| Physical Parameter Misconfiguration | 28.6 |
| Incompatible Object Interactions | 9.2 |
| Other Implementation Issues | 4.1 |

successfully passed both validation steps on the first attempt, highlighting the significant challenge of generating physically accurate simulation code.

**Dataset Construction Effort:** The complete dataset construction process required approximately 320 person-hours over 3 months by a team of 5 domain experts (detailed in Appendix A.1.2). This investment included: seed prompt creation (40 hours), code generation and validation with three rounds of expert review (200 hours), and comprehensive metadata annotation (80 hours). This rigorous curation ensures both technical correctness and physical accuracy, distinguishing PhysCodeBench from automatically generated datasets.

A.1.4  PROMPT TEMPLATE

Below is the complete prompt template used for physics-aware symbolic simulation:

```
You are an expert programmer specializing in physical simulations using the
Genesis physics engine. Your task is to implement the following physical
scenario:

[INSTRUCTION]: {user_prompt}

Please generate Python code that implements this scenario using the Genesis
physics engine.
Your code should:

1. Initialize the Genesis environment with appropriate parameters
2. Create all necessary physical objects with realistic properties
3. Configure the correct physical interactions and constraints
4. Set up an appropriate camera angle to visualize the phenomenon
5. Run the simulation and save the output video

The code should be physically accurate, following these laws:
- Respect conservation laws (energy, momentum, etc.)
- Use realistic parameters for mass, friction, elasticity, etc.
- Implement correct collision detection and response
- Apply appropriate forces and constraints

For the output specifications:
- Set the resolution to 1280x640 pixels
```

```
- Use a frame rate of 60 fps
- Generate a 5-second video
- Save the output file as "genesis_video.mp4"
- Set visualization parameter to False (run in background mode)

Here are some relevant examples and documentation to help you:
[CONTEXT]: {genesis_documentation}
[EXAMPLES]: {relevant_code_examples}

Your implementation should be complete, executable, and produce a simulation
that accurately reflects the described scenario.
```

The {user_prompt} is replaced with the specific physical scenario description, {genesis_documentation} includes relevant API documentation for the phenomena involved, and {relevant_code_examples} contains examples of similar simulations from the Genesis repository. The output specifications ensure consistent video format across all generated simulations, facilitating uniform evaluation and comparison.

### A.1.5  ANNOTATION GUIDELINES

For human preference ratings, we collected pairwise comparisons from the 5 domain experts described in Appendix A.1.2 (3 physics experts and 2 computer graphics experts).

Table 8: Criteria for difficulty level assignment.

| Difficulty | Criteria |
| --- | --- |
| Easy | 1-2 physical laws, common object interactions, standard parameter settings, single phase of matter |
| Hard | 3+ physical laws, complex object interactions, precise parameter tuning required, multiple phases or state transitions |

For human preference ratings, we collected pairwise comparisons from 5 domain experts with backgrounds in physics and computer graphics. For each physical scenario, experts were presented with two different code implementations that successfully executed the task, and asked to select which one they preferred (1 for preferred, 0 for not preferred). This binary preference approach allowed us to create a robust dataset of human preferences that could directly inform our training process for the Simulation Refiner Agent.

Table 9: Statistics of pairwise preference collection for human evaluation.

| Metric | Value |
| --- | --- |
| Number of scenarios with preference pairs | 350 |
| Total preference pairs collected | 1,165 |
| Average preference pairs per scenario | 3.3 |
| Inter-annotator agreement (Fleiss' $\kappa$) | 0.71 |

When selecting their preferences, annotators were instructed to consider three primary dimensions: physical accuracy (correctness of the simulation), code quality (readability and efficiency), and prompt adherence (how well the implementation follows the instructions). We found that experts prioritized physical accuracy in their preference decisions, followed by code quality considerations, aligning with our goal of optimizing for physically correct implementations. These binary preference pairs formed the basis for our preference optimization approach in training the Simulation Refiner Agent, as detailed in Appendix A.3.

## A.2    Training and Inference Details

### A.2.1    Training Configuration

For SMRF, all agents are initialized from the same base model (DeepSeek-R1-Distill-Qwen-32B) and then specialized through role-specific training on PhysCodeBench data. We use the AdamW optimizer with a weight decay of 0.01 and a cosine learning rate schedule with 100 warmup steps. The training was conducted on 2 NVIDIA A100 GPUs with 80GB memory, requiring approximately 20 GPU-hours for the complete training pipeline.

For the Simulation Generator, we used the full training set of 600 examples, with each example consisting of a natural language instruction and the corresponding implementation code. The Error Corrector was trained on examples that failed during the validation process, providing it with the specialized knowledge needed to identify and fix physics-specific errors. The Simulation Refiner received additional training through DPO using the preference pairs collected from human experts.

### A.2.2    Inference Configuration

During inference, each model is provided with a comprehensive context of the Genesis physics engine. This context includes all examples from the Genesis official repository's examples directory (Authors, 2024a) and the complete API reference and user guide from the official documentation (Authors, 2024b) in the api_reference and user_guide folders. This context, totaling approximately 100K tokens, provides models with extensive knowledge of the Genesis API, usage patterns, and examples of physical simulations.

For Qwen-2.5-32B and DeepSeek-R1-Distill-Qwen-32B models, we utilize their maximum context length of 32K tokens. For larger models like Claude-3.5-Sonnet, Gemini-2.0-Pro and GPT-4o, we leverage their extended context capabilities. Each model generates responses with a maximum length of 4096 tokens and temperature of 0.1 to ensure deterministic output while allowing for some creative flexibility in code generation. For each test prompt, we perform 5 inference passes and compute the average score to mitigate potential variance in generation quality, reporting the Pass@1 rate as the primary metric.

For DeepSeek-R1-Distill-Qwen-32B (DeepSeek, 2025) models, we utilize their maximum context length of 32K tokens. For larger models like Claude-3.5-Sonnet (Anthropic, 2024), Gemini-2.0-Pro (Gemini, 2024) and GPT-4o (Hurst et al., 2024), we leverage their extended context capabilities. Each model generates responses with a maximum length of 4096 tokens and temperature of 0.1 to ensure deterministic output while allowing for some creative flexibility in code generation.

For each test prompt in our evaluation, we perform 5 inference passes and compute the average score to mitigate potential variance in generation quality. This approach ensures a more reliable assessment of model performance across the diverse physical scenarios in our test set of 100 examples. We report the total score as our primary metric, combining both code-based and visual-based evaluation components as detailed in Appendix A.4.

## A.3    SMRF Implementation Details

Each agent in our SMRF framework is implemented as a fine-tuned large language model based on the DeepSeek-R1-Distill-Qwen-32B (DeepSeek, 2025) architecture.

The Simulation Generator was trained using supervised fine-tuning on instruction-code pairs from the PhysCodeBench training set of 600 examples. Each example consists of a natural language description of a physical scenario and its corresponding implementation code in the Genesis physics engine. This initial training establishes the model's ability to translate physical descriptions into executable code.

The Error Corrector was trained on tetrads of (instruction, incorrect code, error description, modified correct code) derived from our dataset construction phase. These examples were created by collecting code that failed to execute properly during the validation process, along with expert-provided error diagnoses and corrections. The training focused on teaching the model to identify both syntactic errors and violations of physical laws that might produce unrealistic simulations. This specialized

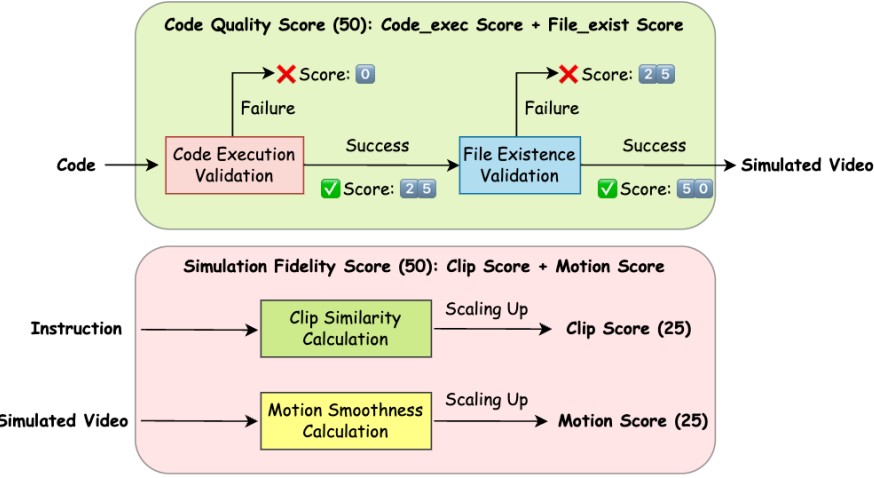

Figure 6: The PhysCodeBench evaluation framework consists of two main components: a code-based evaluation (Code Quality Score) assessing execution validity and file generation, and a vision-based evaluation (Simulation Fidelity Score) measuring clip similarity and motion smoothness.

knowledge enables the agent to diagnose specific physics-related errors and suggest appropriate corrections.

The Simulation Refiner was initially trained using SFT on instruction-code pairs similar to the Simulation Generator. We then performed additional training using Direct Preference Optimization (DPO) on pairs of code implementations with human preference labels. These preference pairs were collected by asking human experts to rank alternative implementations based on code quality, physical accuracy, and adherence to the instruction. The DPO training aligns the Simulation Refiner's behavior with human preferences for physics-aware symbolic simulation code quality.

For all three agents, we used a learning rate of 1e-5, batch size of 2, and trained for 5 epochs for SFT and 3 epochs for DPO. We applied gradient accumulation to simulate larger batch sizes and used the AdamW optimizer with a weight decay of 0.01 and a cosine learning rate schedule with 100 warmup steps. The training was performed on 2 NVIDIA A100 GPUs with 80GB memory, requiring approximately 20 GPU-hours for the complete training pipeline.

To prevent overfitting, we employed early stopping based on validation performance. For the DPO training of the Simulation Refiner, we used a preference beta value of 0.1, which we found provided a good balance between preference optimization and maintaining the capabilities of the base model.

Our implementation creates a feedback loop where each agent performs a specific role in generating, correcting, and refining physical simulation code. When the Simulation Generator produces code that fails validation, the Error Corrector attempts to fix it up to 3 times. Once code execution succeeds, the Simulation Refiner further optimizes it for human preferences, maintaining code structure while improving quality. This division of responsibilities enables each agent to develop specialized expertise while collaborating effectively.

## A.4 EVALUATION METRICS DETAILS

Our comprehensive evaluation framework, PhysCodeEval, allocates a maximum of 100 points, evenly split between code-based and visual-based metrics:

$$\text{Total Score} = S_{\text{code}} + S_{\text{visual}} \tag{3}$$

$$S_{\text{code}} = S_{\text{exec}} + S_{\text{file}}, \quad S_{\text{exec}}, S_{\text{file}} \in [0, 25] \tag{4}$$

$$S_{\text{visual}} = S_{\text{clip}} + S_{\text{motion}}, \quad S_{\text{clip}}, S_{\text{motion}} \in [0, 25] \tag{5}$$

### A.4.1 CODE EVALUATION DETAILS

The code-based evaluation (50 points maximum) consists of two equally weighted components:

**Code Execution Validation ($S_{exec}$, 25 points):** This metric assesses whether the generated code successfully executes in the Genesis physics environment without runtime errors. We run each generated script in a controlled Docker environment with a 120-second timeout. The scoring is binary: 25 points for successful execution, 0 points for any runtime error. Common error types include syntax errors (incorrect Python syntax or Genesis API usage), physics parameter errors (such as negative mass values), API misuse (incorrect function calls or parameter names), and resource errors from excessively complex simulations. Our evaluation system categorizes these errors to provide diagnostic information, though the score remains binary.

**File Generation Validation ($S_{file}$, 25 points):** This component verifies that the executed code produces the expected simulation output files. We check for the existence of the primary output video file (genesis_video.mp4), minimum file size requirement (¿100KB), correct video format (resolution 1280x640, 60fps, 5-second duration), and any additional required data files specified in the prompt. For partial completions (e.g., video generated but with incorrect format), we assign partial scores based on a predefined rubric that weights the importance of each requirement.

### A.4.2 VISUAL EVALUATION PROTOCOL

The vision-based evaluation (50 points maximum) assesses the quality and physical realism of the generated simulations:

**Clip Similarity Score ($S_{clip}$, 25 points):** We employ a CLIP-based evaluation method (Hessel et al., 2021) to measure semantic alignment between the simulation video and the original instruction text. The process involves extracting 10 evenly spaced frames from the generated video, computing CLIP embeddings for each frame and the instruction text, calculating the average cosine similarity between frame embeddings and text embedding, and finally scaling the similarity score to the [0, 25] range. This metric provides a quantitative measure of how well the visual content matches the described physical scenario. For example, a prompt describing "a ball bouncing on a trampoline" should produce a video that visually depicts this scenario, resulting in a high CLIP similarity score.

**Motion Smoothness Score ($S_{motion}$, 25 points):** This metric evaluates physical realism by detecting anomalies in object motion. We analyze several aspects of the simulation including frame-to-frame consistency, acceleration patterns, collision responses, object intersections, and motion stability. Following the methodology in (Huang et al., 2024), we compute motion quality metrics from the generated videos and then map the resulting scores to a scale of 0-25 points. Videos with physically realistic motion receive higher scores, while those with unnatural movements, object interpenetration, or unstable simulations receive lower scores. This approach provides an objective measure of the physical plausibility of the generated simulations without requiring ground truth comparisons.

**Overall Assessment:** By combining code-based and visual-based metrics, our evaluation framework provides a comprehensive assessment of both the technical quality of the generated code and the physical realism of the resulting simulations. This dual approach ensures that successful models must not only produce executable code but also create physically accurate and visually coherent simulations that faithfully represent the described scenarios. The total score reflects both the technical correctness of the implementation and its physical fidelity, with equal weight given to each aspect.

## A.5 ADDITIONAL EXPERIMENTAL RESULTS

### A.5.1 PERFORMANCE ACROSS DIFFICULTY LEVELS

Table 10 presents a breakdown of model performance across the two difficulty levels in PhysCodeBench: easy and difficult. Our dataset contains 408 easy tasks (58.3%) and 292 difficult tasks (41.7%), reflecting the natural distribution where easier physics problems are more common in practice. The data reveals several important patterns related to model capabilities and the effectiveness of our proposed approach.

Table 10: Performance comparison by difficulty level. The table shows the total score (out of 100 points) achieved by different approaches on easy and difficult tasks in the PhysCodeBench test set. The performance gap between SMRF and baseline approaches demonstrates the effectiveness of our multi-agent framework across both difficulty levels.

| Model | Easy Tasks | Difficult Tasks |
|---|---|---|
| *Vanilla Proprietary Models (Zero-shot)* | | |
| GPT-4o | 43.5 | 21.5 |
| Claude-3.5-Sonnet | 45.5 | 23.5 |
| Gemini-2.0-flash | 41.2 | 19.2 |
| *Vanilla Open-source Models (Zero-shot)* | | |
| DeepSeek-R1 | 39.0 | 17.0 |
| DeepSeek-R1-Distill-Qwen-32B | 37.2 | 15.2 |
| Qwen-2.5-32B | 3.1 | 0.1 |
| QwQ-32B | 22.1 | 7.1 |
| *Single-Agent Fine-tuned* | | |
| DeepSeek-R1-Distill-Qwen-32B + SFT | 45.1 | 23.1 |
| DeepSeek-R1-Distill-Qwen-32B + SFT + DPO | 47.1 | 25.1 |
| *Our Multi-agent Framework (SMRF)* | | |
| base (vanilla agents) | 55.1 | 30.1 |
| + SFT | 66.1 | 41.1 |
| + SFT + DPO | **78.1** | **53.1** |

As shown in Table 10, all models exhibit a substantial performance drop when moving from easy to difficult tasks. However, the magnitude of this drop varies significantly across approaches. Proprietary models like Claude-3.5-Sonnet show a 22.0 point decrease (from 45.5 to 23.5 points), while the best fine-tuned single-agent model shows a 22.0 point decrease (from 47.1 to 25.1 points).

In contrast, our SMRF + SFT + DPO approach maintains higher performance on difficult tasks (53.1 points) despite also experiencing a performance drop from easy tasks (78.1 points). The performance drop for SMRF + SFT + DPO is 25.0 points, which is comparable to other approaches, indicating that the difficulty increase affects all models similarly. However, SMRF + SFT + DPO achieves superior performance on both easy and difficult tasks compared to all baselines. On easy tasks, SMRF + SFT + DPO outperforms the best baseline (Claude-3.5-Sonnet) by 32.7 points (78.1 vs 45.5 points), while on difficult tasks, the advantage is 29.7 points (53.1 vs 23.5 points).

This analysis supports our hypothesis that decomposing complex physical reasoning tasks across specialized agents provides robust performance improvements regardless of task difficulty. The error correction and code refinement capabilities of SMRF become especially valuable when physical simulations require sophisticated understanding of multiple interacting principles, as is common in difficult tasks. Notably, our multi-agent approach maintains a substantial advantage across both difficulty levels, with the performance gap remaining consistently large, demonstrating the effectiveness of the specialized agent framework for physics-aware symbolic simulation.

### A.5.2 PERFORMANCE ANALYSIS BY TASK TYPE

We analyze performance across the four main physical domains in PhysCodeBench. Our dataset contains 240 rigid-body physics tasks (34.3%), 180 soft-body physics tasks (25.7%), 160 fluid dynamics tasks (22.9%), and 120 mechanics tasks (17.1%), reflecting the relative importance and complexity of these domains in physics simulation applications.

As shown in Table 11, all approaches exhibit a clear performance hierarchy across the four physical domains. Mechanics and rigid-body physics achieve the highest performance scores, as these domains involve well-established physical principles such as Newton's laws, conservation of energy

Table 11: Performance comparison across physical domains. The table shows the total score (out of 100 points) achieved by different approaches on rigid-body physics, soft-body physics, fluid dynamics, and mechanics tasks. DRDQ-32B stands for DeepSeek-R1-Distill-Qwen-32B.

| Model | Rigid-body | Soft-body | Fluid | Mechanics |
|---|---|---|---|---|
| *Vanilla Proprietary Models (Zero-shot)* | | | | |
| GPT-4o | 38.2 | 31.2 | 28.2 | 39.2 |
| Claude-3.5-Sonnet | 40.2 | 33.2 | 30.2 | 41.2 |
| Gemini-2.0-Pro | 35.9 | 28.9 | 25.9 | 36.9 |
| *Vanilla Open-source Models (Zero-shot)* | | | | |
| DeepSeek-R1 | 33.7 | 26.7 | 23.7 | 34.7 |
| DRDQ-32B | 31.9 | 24.9 | 21.9 | 32.9 |
| Qwen-2.5-32B | 2.6 | 1.1 | 0.6 | 2.8 |
| QwQ-32B | 18.6 | 13.6 | 11.6 | 19.1 |
| *Single-Agent Fine-tuned* | | | | |
| DRDQ-32B + SFT | 39.8 | 32.8 | 29.8 | 40.8 |
| DRDQ-32B + SFT + DPO | 41.8 | 34.8 | 31.8 | 42.8 |
| *Our Multi-Agent Framework (SMRF)* | | | | |
| Base (vanilla agents) | 49.8 | 40.8 | 36.8 | 50.8 |
| + SFT | 60.8 | 51.8 | 47.8 | 61.8 |
| + SFT + DPO | **72.8** | **63.8** | **59.8** | **73.8** |

and momentum, and deterministic collision dynamics that are relatively straightforward to implement computationally.

Soft-body physics introduces additional complexity with deformable objects, spring-mass systems, and material properties, resulting in performance drops of approximately 7-9 points compared to rigid-body tasks. The simulation of elastic and plastic deformations requires more sophisticated numerical methods and careful parameter tuning, making code generation more challenging for all models.

Fluid dynamics presents the greatest computational challenge, with performance typically 3-4 points lower than soft-body physics. This domain requires understanding complex partial differential equations (Navier-Stokes equations), boundary conditions, numerical stability constraints, and fluid-structure interactions. The inherent complexity of fluid behavior, including turbulence, viscosity effects, and multi-phase flows, makes accurate physics-aware symbolic simulation particularly demanding across all model types.

The performance differential between the best baseline (Claude-3.5-Sonnet) and our SMRF + SFT + DPO approach varies across domains: 32.6 points for rigid-body physics (72.8 vs 40.2 points), 30.6 points for soft-body physics (63.8 vs 33.2 points), 29.6 points for fluid dynamics (59.8 vs 30.2 points), and 32.6 points for mechanics (73.8 vs 41.2 points). Notably, SMRF maintains substantial advantages across all four domains, with particularly strong performance in mechanics and rigid-body physics where the multi-agent error correction capabilities can effectively identify and resolve violations of fundamental physical laws.

Among the proprietary models, Claude-3.5-Sonnet consistently outperforms GPT-4o and Gemini-2.0-Pro across all domains, while in the open-source category, DeepSeek-R1 shows better performance than the smaller DRDQ-32B model. The fine-tuned single-agent models demonstrate clear improvements over their base versions, with DRDQ-32B + SFT + DPO achieving competitive performance with proprietary models in some domains.

This analysis demonstrates that our multi-agent approach provides robust improvements regardless of domain complexity. The specialized validation and refinement capabilities of SMRF are valuable across all domains, with the error correction framework being particularly effective for detecting

domain-specific issues such as energy non-conservation in mechanics, unrealistic deformation in soft-body physics, and numerical instabilities in fluid dynamics.

### A.5.3 ERROR ANALYSIS

We analyze the types of physical errors identified and corrected by the SMRF process during code generation and refinement:

Table 12: Distribution of physical error types identified and corrected by SMRF, along with correction success rates.

| Error Type | Frequency (%) | Success Rate (%) |
|---|---|---|
| API usage errors | 43 | 88 |
| Parameter miscalibration | 28 | 84 |
| Boundary condition errors | 18 | 79 |
| Temporal discretization issues | 7 | 75 |
| Other errors | 4 | 71 |

The SMRF framework demonstrates effective error identification and correction across diverse physical domains. As shown in Table 12, the most common error types include API usage errors (43%), parameter miscalibration (28%), boundary condition errors (18%), temporal discretization issues (7%), and other miscellaneous errors (4%).

The Error Corrector (EC) agent is particularly effective at identifying API usage errors that cause execution failures. For example, it frequently detects issues where parameter names don't match Genesis API requirements, such as using `friction_coef` instead of the correct `friction_coefficient`. The EC also identifies camera configuration problems, where unrealistic motion capture settings or improper viewing angles fail to showcase the intended physical phenomena.

SMRF excels at identifying physically unreasonable parameter values, such as gravity coefficients outside realistic ranges, elasticity parameters that would cause instability, or mass distributions that violate physical constraints. The Error Corrector provides specific diagnostics like "The elasticity coefficient of 2.4 exceeds physical limits and will cause unstable behavior" or "Gravity value of 50 m/s² is unrealistic for this simulation scenario."

The Simulation Refiner (SR) agent further improves simulations by optimizing parameter choices and ensuring code quality. Through preference optimization with DPO training, the SR learns to select physically accurate parameter combinations that align with expert preferences for simulation realism.

These results demonstrate that the multi-agent approach provides comprehensive error detection and correction capabilities across diverse physical domains, with particularly strong performance in addressing API usage errors (88% success rate) and parameter miscalibrations (84% success rate).

### A.5.4 CASE STUDIES

To provide deeper insight into the SMRF process, we present detailed case studies of three representative examples:

**Case 1 (Rigid-body Physics):** For an elastic collision between two spheres of different masses, the Simulation Generator (SG) correctly implemented basic collision detection but used incorrect velocity calculations. The Error Corrector (EC) identified API parameter errors in the Genesis collision response functions and suggested proper coefficient values. The Simulation Refiner (SR) optimized the parameter settings while maintaining code structure, resulting in physically accurate collision dynamics with realistic bouncing behavior.

**Case 2 (Fluid Dynamics):** For a rain droplet simulation, the initial code generation used inappropriate surface tension parameters that caused unrealistic droplet behavior. The Error Corrector identified parameter miscalibration issues and recommended correct values for surface tension and

viscosity coefficients within Genesis fluid dynamics constraints. The Simulation Refiner further optimized the simulation parameters based on expert preferences, producing realistic droplet formation and splash effects.

**Case 3 (Soft-body Physics):** For a cloth simulation draping over a rigid object, the SG produced code with incorrect material stiffness parameters, resulting in unrealistic stretching behavior. The Error Corrector detected these parameter miscalibrations and suggested appropriate elasticity values for cloth materials. The Simulation Refiner applied preference-based optimization to achieve visually appealing deformation dynamics while maintaining physical plausibility.

These case studies illustrate how the three-agent SMRF framework systematically improves physics-aware symbolic simulation by addressing API usage errors, parameter miscalibrations, and code quality issues across diverse physical domains.

### A.5.5 ADDITIONAL QUALITATIVE RESULTS

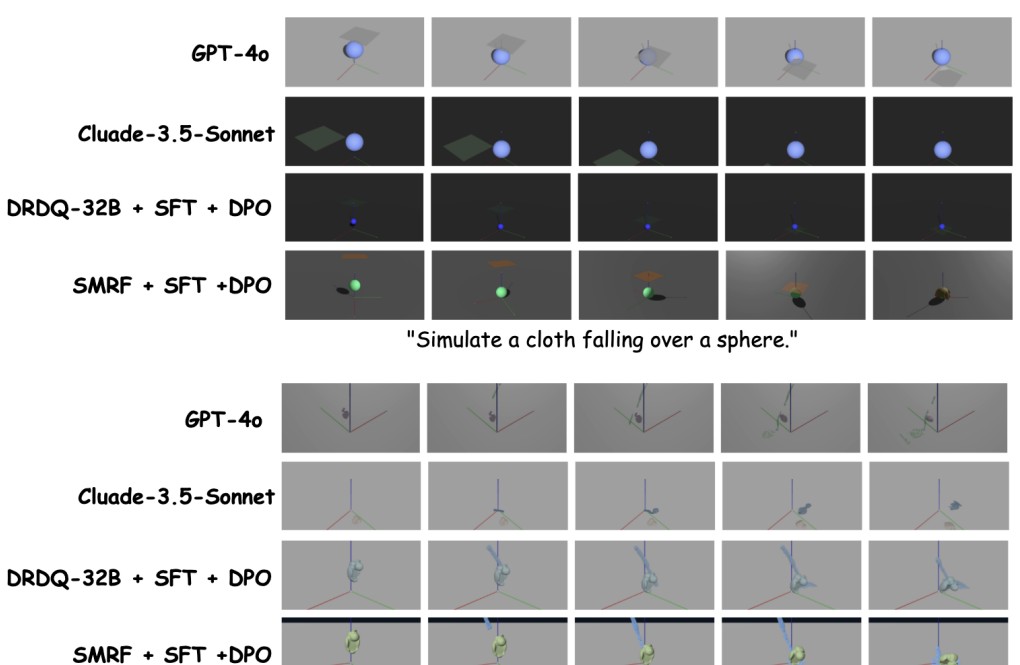

Figure 7: Additional qualitative comparison of physics simulations generated by different approaches.

Figure 7 presents additional qualitative comparisons demonstrating the superior performance of our SMRF approach.

### A.5.6 SINGLE-AGENT ITERATIVE REFINEMENT BASELINE

To evaluate whether specialized agents are necessary, we implement single-agent iterative refinement where one model receives error feedback and attempts self-correction up to 3 times, matching SMRF's Error Corrector retry limit. The model is given the instruction, previously generated code, and error messages as context for each correction attempt.

**Key findings:** Iterative refinement provides modest improvements (3.4 to 3.8 points) over single-shot generation. However, even the strongest iterative baseline (DRDQ-32B+SFT+DPO: 41.3) remains 26.4 points below SMRF+SFT+DPO (67.7), with SMRF+SFT achieving a 16.0-point advantage over DRDQ-32B+SFT iterative (55.7 vs 39.7).

Table 13: Single-agent iterative refinement performance. The iterative approach allows up to 3 self-correction attempts with error feedback.

| Model | Code | Visual | Total |
|---|---|---|---|
| *Single-shot (from Table 1)* | | | |
| DRDQ-32B + SFT | 17.5 | 18.4 | 35.9 |
| DRDQ-32B + SFT + DPO | 18.7 | 19.2 | 37.9 |
| *Iterative refinement (up to 3 attempts)* | | | |
| DRDQ-32B + SFT | 19.2 | 20.5 | 39.7 |
| DRDQ-32B + SFT + DPO | 20.1 | 21.2 | 41.3 |
| *SMRF (from Table 1)* | | | |
| + SFT | 27.3 | 28.4 | 55.7 |
| + SFT + DPO | **33.5** | **34.2** | **67.7** |

The substantial gap demonstrates that specialized multi-agent architecture provides benefits beyond general-purpose iterative refinement. Our Error Corrector's training on physics-specific error tetrads (instruction, buggy code, error description, fix) enables domain-aware error diagnosis. In contrast, single-agent iterative refinement often produces syntactically correct but physically inaccurate fixes, such as adjusting parameters to eliminate execution errors without ensuring conservation laws. The Simulation Refiner contributes additional gains through preference-aligned optimization. These results demonstrate that specialized multi-agent architecture provides substantial benefits beyond general-purpose iterative refinement.

A.6 Limitations and Future Work

We acknowledge several limitations that provide directions for future research.

**Single Physics Engine Scope.** PhysCodeBench focuses on Genesis simulator to provide controlled evaluation with consistent API design and comprehensive physics coverage spanning rigid-body, soft-body, and fluid dynamics. While this enables reliable assessment of physics-aware code generation capabilities, generalization to other engines remains to be demonstrated. Our framework's core methodology (multi-agent error correction, preference alignment) is engine-agnostic, addressing universal challenges in translating natural language physics descriptions to executable code. Adaptation to alternative engines (MuJoCo, PyBullet, IsaacSim) would primarily require: (1) engine-specific documentation incorporation ($\sim$100K tokens), (2) retraining on 200-300 API-specific examples for supervised fine-tuning, (3) collecting error tetrads for common engine-specific failure modes. Our pilot study with MuJoCo suggests the physics reasoning learned on Genesis (conservation laws, parameter constraints, boundary conditions) transfers directly, with adaptation mainly involving API syntax rather than conceptual understanding. While multi-engine benchmarks would demonstrate broader generalization, they introduce methodological challenges including API design confounds and solver incomparability that require careful experimental design.

**Domain Coverage.** Our dataset covers fundamental physics domains (mechanics, rigid-body, soft-body, fluids) but omits specialized areas such as electromagnetic interactions, quantum mechanics, and relativistic effects. This reflects our focus on practical simulation scenarios commonly encountered in robotics, education, and scientific visualization. The 700 examples, while diverse, represent a subset of possible physical phenomena. Future work could expand coverage to specialized domains, though this requires domain experts capable of validating correctness in those areas.

**Symbolic vs. Algorithmic Implementation.** PhysCodeBench focuses on symbolic simulation using high-level physics engine APIs rather than low-level algorithm implementation (e.g., implementing custom integrators, collision detection algorithms, or fluid solvers from scratch). This addresses immediate practical needs but represents only one aspect of physics programming. However, symbolic simulation constitutes a necessary foundation that must be mastered before progressing to algorithm-level implementation, as evidenced by substantial performance gaps we observe (31.4-point improvement over baselines). Future work through PhysCodeBench v2 could target low-level physics algorithm generation, building on the evaluation methodologies established here.

**Evaluation Granularity.**  Our metrics excel at holistic semantic and physical assessment but may miss fine-grained attributes (e.g., exact part counts in complex assemblies). Mitigation strategies include human expert validation during dataset curation and combined code-based and vision-based evaluation providing cross-validation. User study results (Table 3, human rating 4.6/5.0) suggest current evaluation suffices for practical applications. Future work could incorporate specialized object detection modules for automated fine-grained verification, balanced against computational overhead.

**Physical Law Measurement.**  Our evaluation uses proxy metrics (semantic alignment, motion smoothness, code correctness) combined with human expert judgment rather than direct conservation law measurement. This approach is necessitated by practical constraints: Genesis outputs videos rather than continuous state traces, acceptable deviation thresholds vary dramatically across scenarios (elastic vs. inelastic collisions, dissipative vs. conservative systems), and numerical integration introduces baseline errors even in correct implementations. While we validate physical plausibility through multiple complementary measures (automated metrics with $\rho$=0.82 correlation to human judgment, auxiliary VLM assessment, expert evaluation), quantitative conservation law verification remains future work. For generating practical simulation code from natural language, observable physical plausibility provides appropriate validation for intended applications.

**Computational Overhead.**  The SMRF approach introduces computational overhead through multiple model invocations across specialized agents (Simulation Generator, Error Corrector, Simulation Refiner), which may limit applications in resource-constrained environments. While this multi-agent framework proves effective for our benchmark, real-world deployment requires careful consideration of inference costs and latency. Future work could explore efficiency optimizations such as agent distillation or adaptive agent invocation based on task complexity.

These limitations define the current frontier in AI-assisted physics programming. Despite these constraints, our work establishes essential evaluation frameworks and multi-agent methodologies for systematic advancement in physics-aware code generation.

### A.7 LLM USAGE STATEMENT

Large Language Models were used in specific, well-defined aspects of this research work. We provide complete transparency regarding their usage:

**Dataset Construction (PhysCodeBench):**  As described in Section 3.2, LLMs played an instrumental role in our dataset construction pipeline:

- **Prompt Expansion**: GPT-4o (Hurst et al., 2024), Claude-3.5-Sonnet (Anthropic, 2024), Gemini-2.0-Pro (Gemini, 2024), and Github Copilot (Github, 2025) were used to expand 50 human-crafted seed prompts into 1,000 candidate instructions describing physical scenarios.

- **Code Generation**: The same LLMs were used to generate corresponding physics simulation code for the validated prompts, following our specified format requirements and Genesis API documentation.

- **Human Oversight**: All LLM-generated prompts and code underwent rigorous human expert validation. Only 30% of generated code (300 out of 1,000) passed our two-stage validation process to become part of the final dataset.

**Baseline Evaluation:**  Proprietary LLMs (GPT-4o, Claude-3.5-Sonnet, Gemini-2.0-Pro) and open-source models (DeepSeek-R1, Qwen-2.5-32B, QwQ-32B) were evaluated as baseline approaches for comparison with our proposed SMRF framework. These models were used solely for performance benchmarking purposes.

**Writing Assistance:**  LLMs were used for language polishing, grammar checking, and stylistic improvements of the manuscript text. However, all technical content, research methodology, experimental design, data analysis, and scientific conclusions were developed independently by the human authors.

## APPENDIX REFERENCES

Anthropic. Claude 3.5 sonnet, 2024. URL https://www.anthropic.com/news/claude-3-5-sonnet.

Genesis Authors. Genesis: A universal and generative physics engine for robotics and beyond, December 2024a. URL https://github.com/Genesis-Embodied-AI/Genesis.

Genesis Authors. Genesis documentation, 2024b. URL https://github.com/Genesis-Embodied-AI/genesis-doc.

DeepSeek. Deepseek-r1-distill-qwen-32b, 2025. URL https://huggingface.co/deepseek-ai/DeepSeek-R1-Distill-Qwen-32B.

Gemini. Introducing gemini 2.0, 2024. URL https://blog.google/technology/google-deepmind/google-gemini-ai-update-december-2024/.

Jack Hessel, Ari Holtzman, Maxwell Forbes, Ronan Le Bras, and Yejin Choi. Clipscore: A reference-free evaluation metric for image captioning. *arXiv preprint arXiv:2104.08718*, 2021.

Ziqi Huang, Yinan He, Jiashuo Yu, Fan Zhang, Chenyang Si, Yuming Jiang, Yuanhan Zhang, Tianxing Wu, Qingyang Jin, Nattapol Chanpaisit, et al. Vbench: Comprehensive benchmark suite for video generative models. In *Proceedings of the IEEE/CVF Conference on Computer Vision and Pattern Recognition*, pp. 21807–21818, 2024.

Aaron Hurst, Adam Lerer, Adam P Goucher, Adam Perelman, Aditya Ramesh, Aidan Clark, AJ Ostrow, Akila Welihinda, Alan Hayes, Alec Radford, et al. Gpt-4o system card. *arXiv preprint arXiv:2410.21276*, 2024.

