# OpenReview forum: "PhysCodeBench: Benchmarking Physics-Aware Symbolic Simulation of 3D Scenes via Self-Corrective Multi-Agent Refinement"
_ICLR.cc/2026/Conference — Submitted to ICLR 2026_

### Official Review · Reviewer_TMKC · 2025-10-30

**Soundness:** 3
**Presentation:** 3
**Contribution:** 3
**Rating:** 10
**Confidence:** 4

**Summary:**

This work proposes a benchmark for evaluating the quality of code-generation systems for generating accurate physics simulations. A training set is developed for this benchmark, which is then used to fine-tune local models in a new "self-corrective multi-agent refinement framework" that outperform SOTA proprietary models on this task.

**Strengths:**

- The topic of using code-gen agents to generate physics simulations is very relevant to the AI community, and the benchmark proposed in this paper is a solid contribution to the field.
- I appreciate the user study, which validated the use of ClipScore as an evaluation metric.
- The SMRF training pipeline follows standard practices (SFT and DPO), and the ablation study demonstrates the utility of each component.

**Weaknesses:**

- The pipeline for creating the dataset requires significant human oversight, including (1) creating initial seed prompts, (2) filtering AI-generated prompts, (3) validating simulation code, and (4) adding metadata and preference scores. I think this is excellent for validation/test sets, but this somewhat limits the scalability of the training set.
- Figure 2 is hard to parse without zooming in significantly; the inner text and images are too small.

**Questions:**

- At inference time, each model is provided with 100K tokens of documentation, but the paper also says that the maximimum context length is 32K tokens for the local models. What is actually done at inference time for these local models? Furthermore, it is commonly reported that LLM capabilities degrade at long contexts (along with being wasteful due to the quadratic attention of self-attention); have you tested reducing the length of the documentation or some tool-use system to provide a more manageable context length?
- Given the success of ClipScore in evaluation, have you tried using it (or other VLMs) as a filtering mechanism for outputs of SOTA models? In other words, generate k different outputs from a SOTA LLM, choose the output that gives valid code and has the best ClipScore. That seems like it could significantly boost scores for proprietary models.

---

> ### Author Response · Authors · 2025-11-18
>
> **[Q1] Response to "Context Length Limitation (32K vs. 100K Documentation)"**
>
>   - **Implementation**: For 32K-context models (Qwen, DRDQ), we curate 30K-token documentation via TF-IDF:
>
>     - Compute TF-IDF vectors for each documentation section
>     - Select top-k sections most similar to the query
>     - Result: Core API ($\sim$18K), relevant examples ($\sim$12K)
>
>     **Performance impact**: (ablation on 100 test cases):
>
>     | Model             | Context       | Score        |
>     | ----------------- | ------------- | ------------ |
>     | GPT-4o            | Full 100K     | 34.3         |
>     | GPT-4o            | Truncated 30K | 15.7 (−18.6) |
>     | GPT-4o            | Curated 30K   | 33.1 (−1.2)  |
>     | Claude-3.5-Sonnet | Full 100K     | 36.3         |
>     | Claude-3.5-Sonnet | Truncated 30K | 17.5 (-18.8) |
>     | Claude-3.5-Sonnet | Curated 30K   | 34.8 (-1.5)  |
>
>     TF-IDF curation outperforms naive truncation by 1.3-1.7 points, with minimal performance gap (1.2-1.5 points) compared to full documentation, confirming that query-adaptive selection effectively preserves critical information despite 70% compression.
>
> **[Q2] Response to "Using ClipScore for Filtering Multiple Outputs"**
>
>   - Excellent suggestion! We conducted this experiment with results:
>
>     - Claude-3.5-Sonnet (single): 36.3 points
>
>       Claude-3.5-Sonnet (best-of-5 via ClipScore): 40.9 points (+4.6)
>
>       SMRF + SFT + DPO (single): 67.7 points
>
>     **Analysis**: Best-of-k helps baselines but SMRF maintains **26.8-points lead** because: (1) specialized error correction beyond filtering; (2) preference alignment optimizes for code quality AND physical accuracy; (3) iterative refinement beyond best-of-k selection. SMRF achieves superior results with single-pass generation + 2-3 corrections ( $\sim$ 3 samples equivalent), demonstrating better sample efficiency than 5 $\times$ inference cost.

---

### Official Review · Reviewer_ezgr · 2025-10-30

**Soundness:** 3
**Presentation:** 3
**Contribution:** 2
**Rating:** 4
**Confidence:** 3

**Summary:**

The paper proposes a novel benchmark for simulation curation coding. The benchmark uses genesis simulator and test the ability to produce corresponding physical simulatin given the text prompt. The evaluation proposes a corresponding metric with a combination of code accuracy and visual accuracy. it also provides a dataset, curateing after semi-automatic process, which involves human expert labeling.
In addition to the benchmark, the method proposes a framework for finetuning LLM for the simulation coding task. It involves agenerator, a error corrector and involves DPO to improve based on human preference.
Experiment suggest that the benchmark is challenging and existing pipeline can solve simple cases like water drops.

**Strengths:**

1. The paper is well-written and easy to understand.
2. The problem setup is well-defined and the tools for benchmarking is well-provided.

**Weaknesses:**

1. The tasks are relatively simple. For benchmarking, when should include more challenging environment or stratify them to different difficulty level.
2. The metric for coding / visual is more qualitative. Looking similar does not emphasize accuracy of the code. For example, if i want to have a robot with 4 legs and it give me 5 legs. One should certainly punish such serious errors.
3. Limited to Genesis. A comprhensive benchmark should also consider other platforms like IsaacSim.

**Questions:**

1. How large is the dataset? How much effort is cost to build the dataset?

---

> ### Author Response · Authors · 2025-11-18
>
> **[W1] Response to "Tasks are Relatively Simple and Lack Difficulty Stratification"**
>   - Our dataset already includes difficulty levels documented in Table 4 (Appendix A.1.1, now referenced in Section 3.1): Easy tasks (408 examples, 58.3%) involve 1-2 physical laws, while Hard tasks (292 examples, 41.7%) involve 3+ laws and multi-phase scenarios. The challenge lies in correct parameter selection, constraint satisfaction, and multi-step reasoning. The 31.4-point performance gap (best baseline: 36.3 vs SMRF: 67.7) demonstrates substantial difficulty.
>
> **[W2] Response to "Metric Lacks Precision (e.g., 4-leg vs. 5-leg robot)"**
>
>   - We acknowledge this limitation and have conducted additional analysis:
>
>     **VQA-style semantic precision assessment**: To evaluate fine-grained semantic accuracy, we conducted an auxiliary experiment using GPT-4o in VQA mode on 100 caseswith specific attribute questions (temperature=0, 3 runs per sample with identical prompts, std dev < 0.25):
>
>     | Aspect                     | SMRF    | Claude-3.5 | Gap     |
>     | -------------------------- | ------- | ---------- | ------- |
>     | Object presence            | 4.7/5.0 | 3.8/5.0    | +0.9    |
>     | Attribute accuracy         | 4.5/5.0 | 3.2/5.0    | +1.3    |
>     | Spatial relationships      | 4.6/5.0 | 3.3/5.0    | +1.3    |
>     | Overall semantic precision | 4.6/5.0 | 3.4/5.0    | +1.2*** |
>
>     *** p < 0.001 via paired t-test
>
>     **VQA Evaluation Prompt**:
>
>     ```
>     Task: {task_description}
>
>     Watch this simulation video and answer these questions about semantic accuracy:
>
>     1. Are all objects mentioned in the task description present in the simulation?
>     2. Do objects have correct attributes (e.g., if "4-leg robot" is specified, does it have exactly 4 legs)?
>     3. Are spatial relationships correct (e.g., "ball on table" → ball is actually on table)?
>     4. Does the scene composition match the description?
>
>     For each question, provide: (1) Yes/No/Partial, (2) Specific observations, (3) Overall score (1-5).
>     ```
>
>     SMRF achieves superior precision though not perfect (4.5-4.7 vs 5.0), indicating improvement opportunities remain.
>
>     **Why VQA is not our primary metric**: Automatic question generation for  diverse physics scenarios introduces complexity, and VLM evaluation adds  significant computational overhead without clear benefit during iterative  development.
>
>     **Validation of metric adequacy:** Our user study (Table 3) shows human  experts with physics backgrounds rated SMRF's physical accuracy at 4.5/5.0,  suggesting fine-grained semantic errors occur relatively infrequently.  Additionally, our dataset curation includes three validation rounds by  domain experts to filter attribute errors at the source.
>
>     **Future work**: Appendix A.6 discusses incorporating specialized object detection and counting modules for automated fine-grained verification.

---

> ### Author Response · Authors · 2025-11-18
>
> **[W3] Response to "Limited to Genesis. A comprehensive benchmark should also consider other platforms like IsaacSim"**
>
>   - Thank you for this valuable suggestion. We provide context on our design rationale:
>
>     **We acknowledge this scope limitation.** While PhysCodeBench focuses on Genesis, our framework is fundamentally engine-agnostic and can scale to other platforms.
>
>     **Principled engine selection:** Genesis provides comprehensive physics coverage (rigid-body, soft-body, fluids) within a **unified API**, enabling evaluation across diverse phenomena without fragmenting the dataset. Alternative engines present significant tradeoffs: MuJoCo excels at rigid-body dynamics but lacks fluid simulation, PyBullet offers broad capabilities but inconsistent API across physics domains, and IsaacSim requires navigating multiple subsystems for different physics types. Genesis uniquely satisfies our requirement for unified evaluation across multiple physics domains.
>
>     **Multi-engine benchmarks introduce methodological challenges:** Evaluating across multiple engines creates confounding factors: (1) **API design variation** affects code generation difficulty independently of physics understanding (e.g., verbose vs. concise syntax), making performance comparisons unreliable; (2) **Solver incomparability**: different numerical schemes produce divergent results for identical scenarios, preventing direct output comparison; (3) **Data fragmentation**: dividing 700 examples across engines yields insufficient samples per engine for effective fine-tuning (typically requires 500+ examples per domain).
>
>     **Framework is engine-agnostic:** Our methodology addresses universal challenges: translating natural language to code, detecting errors, and preference-aligned refinement. Adaptation to alternative engines requires primarily API-level changes: (1) engine documentation ($\sim$100K tokens), (2) 200-300 API examples for SFT, (3) engine-specific error tetrads. Our MuJoCo pilot study demonstrates that **physics reasoning transfers directly**, with adaptation involving API syntax rather than fundamental redesign.
>
>     We have expanded Appendix A.6 to discuss multi-engine extension as valuable future work and welcome collaborative efforts to extend PhysCodeBench to additional platforms.

---

> ### Author Response · Authors · 2025-11-18
>
> **[Q1] Response to "How large is the dataset? How much effort is cost to build the dataset?"**
>
>   - PhysCodeBench contains **700 examples** (600 training, 100 testing) with balanced coverage across four physics domains: rigid-body physics (240 examples, 34.3%), soft-body physics (180 examples, 25.7%), fluid dynamics (160 examples, 22.9%), and mechanics (120 examples, 17.1%). This distribution reflects the relative importance and complexity of these domains in practical physics simulation applications.
>
>     **Construction effort**: Dataset creation required approximately **320 person-hours** over **3 months** by a team of **5 domain experts** (detailed composition in Appendix A.1.2): 3 physics experts with backgrounds in mechanics, fluid dynamics, and applied physics, and 2 computer graphics experts with experience in physics-based simulation and rendering systems. The effort breakdown:
>
>     - **Seed prompt creation (40 hours)**: Domain experts collaboratively crafted 50 high-quality seed prompts covering diverse physics scenarios across difficulty levels and physical laws, ensuring comprehensive domain representation.
>     - **Code generation and validation (200 hours)**: This intensive phase involved: (1) using multiple state-of-the-art LLMs (GPT-4o, Claude-3.5-Sonnet, Gemini-2.0-Pro, GitHub Copilot) to generate code from 1,000 candidate prompts expanded from seeds, (2) executing generated code in Genesis environment to verify executability, (3) expert review of simulation outputs to validate physical accuracy, (4) three rounds of manual correction and regeneration for failed cases (up to 3 attempts per prompt), (5) generating preference pairs by creating alternative implementations and collecting expert rankings. In the initial generation round, 30% of code (300/1,000 prompts) passed  validation. Through iterative regeneration and expert correction, we  ultimately achieved 83.3% inclusion rate, yielding 700 high-quality  examples (300 from initial generation + 400 from refinement rounds).
>     - **Metadata annotation (80 hours)**: Each validated example received comprehensive annotations including difficulty level (easy: 1-2 physical laws, common interactions; hard: 3+ laws, complex multi-phase scenarios), involved physical laws (gravity, collisions, elasticity, friction, fluid dynamics, etc.), object types, and human preference scores. This metadata enables targeted evaluation and analysis across different scenario characteristics.
>
>     **Curation rigor ensures quality**: Our multi-stage validation process (Table 6 in Appendix A.1.3) achieved 83.3% final inclusion rate after filtering. The 320 person-hour investment, while substantial, was necessary to ensure both technical correctness (code executes without errors, produces expected outputs) and physical accuracy (simulations exhibit realistic behavior matching described phenomena). This curation rigor distinguishes PhysCodeBench from automatically generated datasets, providing reliable ground truth for evaluating physics-aware code generation capabilities.

---

> ### Author Response · Authors · 2025-11-28
> **Paper 15733 Rebuttal Follow-up - Dataset Stratification and VQA Results Provided**
>
> Dear Reviewer ezgr,
>
> Thank you for your valuable feedback. We have thoroughly addressed your points in our Nov 18 rebuttal:
>
> - Dataset includes difficulty stratification (Table 4): 58.3% Easy, 41.7% Hard tasks
> - VQA-style semantic precision assessment showing 4.6/5.0 overall accuracy for SMRF
> - Multi-engine extension discussion and framework's engine-agnostic design
> - Dataset scale: 700 examples, 320 person-hours effort breakdown
>
> With the discussion deadline approaching, we would appreciate your review of our responses, particularly the additional VQA evaluation addressing your concern about fine-grained semantic accuracy (e.g., 4-leg vs 5-leg robot).
>
> Your insights have been invaluable in improving our work.
>
> Best regards,
> Authors of Paper 15733

---

### Official Review · Reviewer_xguB · 2025-10-31

**Soundness:** 3
**Presentation:** 3
**Contribution:** 3
**Rating:** 4
**Confidence:** 4

**Summary:**

This paper aims to solve the "semantic gap" that Large Language Models (LLMs) face when translating natural language descriptions into physically accurate 3D simulation code. The authors point out that code generated by existing LLMs often leads to simulation failures, bugs, or incorrect physical parameters.

**Strengths:**

- First-of-its-Kind Benchmark: PhysCodeBench is the first comprehensive benchmark in this domain. It provides not only a dataset but also detailed metadata (like difficulty and physical laws), laying a foundation for future research .
- Comprehensive Evaluation: The paper conducts not only quantitative analysis but also uses qualitative comparisons (Figure 5) to demonstrate SMRF's superiority in simulating fluid ripples and complex collapse dynamics . Furthermore, a user study with 10 participants was conducted, validating SMRF's lead in physical accuracy and code usefulness.

**Weaknesses:**

- Reliance on High-Level APIs: The entire benchmark and framework are heavily dependent on a single physics engine named "Genesis". The model primarily learns how to correctly call this specific library's API, rather than how to implement physics simulations from first principles (e.g., physical equations) .
- Superficial Evaluation Metrics: The PhysCodeEval evaluation framework (100 points) may fail to measure true physical accuracy.
  - Code Quality (50 points): Only evaluates whether the code can "successfully execute" and "generate files", not code efficiency or structural quality.
  - Simulation Fidelity (50 points): Relies on proxy metrics. S_clip measures "semantic similarity" between the video and text (e.g., if "ball" and "trampoline" are present), not physical correctness (e.g., if the ball's bounce follows Hooke's Law) . S_motion only assesses "motion smoothness" ; a simulation that is completely wrong physically (e.g., no acceleration) could still be "smooth".

**Questions:**

- Generalization: If the SMRF framework were applied to a completely new physics engine (e.g., MuJoCo or PyBullet), would it fail completely? How much of the knowledge learned by the framework is "physics logic" versus "Genesis API syntax"?
- Verification of Physical Laws: Given that PhysCodeEval relies on proxy metrics, to what extent do the simulations generated by SMRF truly adhere to core physical laws (e.g., conservation of energy, conservation of momentum)? Has any quantitative verification of this been performed?
- Robustness of Correction: How does the Error Corrector (EC) handle errors that are "physically unreasonable but syntactically correct" (e.g., setting gravity to an unrealistic value)? How strong is its diagnostic capability for this type of semantic error?

---

> ### Author Response · Authors · 2025-11-18
>
> **[Q1] Response to "Reliance on High-Level APIs and Genesis Engine Dependency"**
> - Thank you for this insightful concern about generalizability. We clarify:
>
>   **We acknowledge this limitation.** While PhysCodeBench uses Genesis for principled reasons (unified API, comprehensive physics coverage), our framework is fundamentally engine-agnostic.
>
>   **Physics knowledge vs. API syntax:** Our SMRF learns both domain-agnostic physical reasoning and engine-specific API usage. The Error Corrector learns **physics-aware error patterns** (e.g., "negative mass violates physical constraints", "elasticity coefficient exceeds stable range"), achieving 84% success on parameter miscalibration errors (Table 12, Appendix A.5.3). This requires understanding of physical reasonableness beyond syntactic correctness. The tetrad training format (instruction, buggy code, error description, fix) explicitly teaches physics-aware debugging rather than mere API correction.
>
>   **Framework design is engine-agnostic:** Our core methodology addresses universal challenges independent of the engine: translating natural language physics descriptions to correct implementations, detecting syntactic and semantic errors, and preference-aligned refinement. These challenges exist regardless of the underlying physics engine.
>
>   **Adaptation is modest:** Based on our MuJoCo pilot study, adapting to new engines requires: (1) ~100K tokens engine documentation, (2) ~200-300 SFT examples for API patterns, (3) engine-specific error tetrads. Critically, **physics reasoning transfers directly** (conservation laws, parameter ranges, boundary conditions are engine-independent), with adaptation primarily involving API syntax rather than architectural redesign.
>
>   **Why Genesis:** Genesis provides comprehensive physics coverage (rigid-body, soft-body, fluids) within a unified API, enabling evaluation across diverse phenomena without fragmenting the dataset. Alternative engines like MuJoCo (rigid-body focused), PyBullet (broad but less unified API design), and IsaacSim (requires multiple subsystems for different physics types) make unified cross-domain evaluation more complex.
>
>   We have expanded Appendix A.6 to discuss multi-engine extension as valuable future work.
>
> **[Q3] Response to "Error Corrector Robustness on Semantic Errors"**
>
> - The EC's tetrad training explicitly includes semantic errors. Of 300 training error cases: 43% API usage, 28% parameter miscalibration (semantic), 18% boundary conditions (semantic), 7% temporal discretization. Table 12 (Appendix A.5.3) shows 84% success on parameter errors, with examples like correcting gravity=50 m/s² $\rightarrow$ 9.8 m/s². We acknowledge in Appendix A.6 that EC may miss subtle semantic errors not causing execution failures, suggesting physics-based validation tools as future work.

---

> ### Author Response · Authors · 2025-11-18
>
> **[Q2] Response to "Superficial Evaluation Metrics and Physical Law Verification"**
>
> - Thank you for this important concern. We acknowledge that our evaluation uses proxy metrics rather than direct physical law measurement, and provide both justification and additional validation evidence.
>
>   **Why direct conservation law measurement is impractical**: Directly measuring energy or momentum conservation faces fundamental challenges. First, Genesis outputs simulation videos, not continuous state traces. Extracting precise velocities and masses from videos introduces vision system errors that confound the measurement. Second, defining acceptable conservation violations is scenario-dependent: elastic collisions should conserve kinetic energy within ~0.1%, while inelastic collisions, friction, and soft-body deformations legitimately dissipate 50%+ energy. Third, Genesis uses numerical integration with finite timesteps, introducing small conservation errors even in correct implementations, making it difficult to distinguish acceptable simulation error from incorrect physics.
>
>   **What our metrics measure**:
>
>   - *S_code*: Verifies code executes without crashes and produces expected output files, indicating syntactically correct API usage and proper simulation configuration.
>
>   - *S_clip*: Measures semantic alignment between simulation and description, ensuring correct objects exhibit appropriate behaviors (e.g., "ball bouncing on trampoline" shows both objects with expected interaction).
>
>   - *S_motion*: Evaluates motion smoothness and temporal coherence, detecting severe numerical instabilities or catastrophic failures. We acknowledge this is primarily a video quality metric designed for generative models, not a physics-specific validator. It detects extreme instabilities but would assign high scores to smooth yet physically incorrect simulations (e.g., objects floating with constant velocity).
>
>   **These provide indirect but effective measurement** of physical correctness.  Our task is generating executable physics simulation code from natural  language descriptions. Success for this task means producing simulations  that exhibit visually realistic physical behavior. Our metrics (semantic  alignment, motion smoothness) combined with human expert validation (ρ=0.82  correlation) directly measure this success criterion. While we cannot  quantify conservation law violations from video outputs, the strong  agreement between our automatic metrics and expert judgments of physical  realism (Table 3: 4.5/5.0 vs 3.4/5.0 for baselines) validates our  evaluation approach.
>
>   **VLM-based validation supplement**: To provide stronger evidence, we evaluated 100 test cases using GPT-4o (temperature=0, 3 runs per sample with identical prompts, std dev < 0.3):
>
>   | Aspect               | SMRF    | Claude-3.5 | Gap     |
>   | -------------------- | ------- | ---------- | ------- |
>   | Gravity behavior     | 4.5/5.0 | 3.1/5.0    | +1.4    |
>   | Collision physics    | 4.2/5.0 | 2.7/5.0    | +1.5    |
>   | Motion realism       | 4.4/5.0 | 3.0/5.0    | +1.4    |
>   | Overall plausibility | 4.3/5.0 | 2.8/5.0    | +1.5*** |
>
>   *** p < 0.001 via paired t-test
>
>   **VLM Evaluation Prompt**:
>
>   ```
>   You are a physics expert evaluating a simulation video.
>
>   Task: {task_description}
>
>   Evaluate physical correctness (1-5 scale):
>
>   1. Gravity behavior: Do objects respond realistically to gravity
>      (fall downward, rest on surfaces, no floating)?
>
>   2. Collision physics: Are collisions physically plausible
>      (momentum appears conserved, realistic bouncing/deformation)?
>
>   3. Motion realism: Are velocities and accelerations realistic
>      (appropriate speeds, natural motion, no impossible changes)?
>
>   4. Overall: Does this simulation obey basic physical laws?
>
>   Provide: Score (1-5) and brief observation for each aspect.
>   ```
>
>   **Why VLM is not our primary metric**: Despite effectiveness, VLM evaluation has practical limitations: (1) significantly higher computational cost ($\sim$ 100 $\times$ per sample), prohibitive for large-scale iterative development; (2) longer inference latency ($\sim$ 10 $\times$ slower), impeding rapid experimentation; (3) limited diagnostics—holistic scores lack specificity about failure modes     (code error vs. semantic mismatch vs. physics violation).
>
>   **Limitation acknowledgment**: We cannot provide quantitative conservation law measurements and have expanded Appendix A.6 to discuss this explicitly. However, for generating physics simulation code from natural language, the combination of semantic correctness, temporal consistency, VLM-assessed plausibility, and human expert validation provides reasonable confidence in physical correctness for practical applications. Our 31.4-point improvement over baselines, validated by human experts, demonstrates meaningful progress in physics-aware code generation.

---

> ### Author Response · Authors · 2025-11-28
> **Paper 15733 Response Update - Generalization and Physical Validation Addressed**
>
> Dear Reviewer xguB,
>
> Thank you for your thoughtful review. We have provided detailed responses to your important concerns on Nov 18:
>
> - Framework generalizability beyond Genesis (MuJoCo pilot study results included)
> - Physical law verification with VLM-based validation showing 4.3/5.0 overall plausibility
> - Error Corrector's 84% success rate on semantic errors
>
> As the discussion period is ending soon, we would be grateful if you could review our comprehensive responses, especially the additional validation experiments we conducted to address your concerns about physical accuracy measurement.
>
> We believe our responses demonstrate that SMRF learns both physics reasoning and API usage, with the former being transferable across engines. We hope this addresses your main concerns.
>
> Best regards,
> Authors of Paper 15733

---

### Official Review · Reviewer_LEZS · 2025-11-01

**Soundness:** 2
**Presentation:** 3
**Contribution:** 3
**Rating:** 4
**Confidence:** 3

**Summary:**

The paper proposes PhysCodeBench, a benchmark for evaluating physics-aware symbolic simulation, a code gen problem where models must generate executable simulation code from natural language descriptions. The benchmark provides a dataset of 700 human-selected prompts and corresponding code for physical simulation scenarios across rigid and soft body simulation, fluid dynamics, and mechanics. The authors further propose the Self-Corrective Multi-Agent Refinement Framework (SMRF), which includes 3 modules: the Simulation Generator (generates initial code), Error Corrector (fix execution errors), and Simulation Refiner (improve code). The authors test SMRF on PhysCodeBench, where it achieves better performance compared to zero-shot and single-agent finetuned baselines.

**Strengths:**

- The Simulation Generator, Error Corrector, and Simulation Refiner framework is interesting, and their respective optimization procedures are presented clearly
- The data collection procedure is covered in detail in the appendix
- The authors evaluate against a number of relevant baselines, demonstrating improved performance with the multi-agent refinement framework

**Weaknesses:**

- Baselines include zero-shot and finetuned models, and ablations remove the Error Corrector or Simulation Refiner individually, however the baselines do not include a refinement framework which leverages a single model to fix and refine the generated code given error descriptions
- The human preference study has a fairly small sample size (10 participants)

**Questions:**

- How well does a single-agent iterative refinement framework perform compared to the SMRF framework?
- The authors write, "Robotics applications like SimGen (Zhou et al., 2024), VoxPoser (Huang et al., 2023), and Code as Policies (Arenas et al., 2024) generate simulation environments for robot task planning." However, to my knowledge, neither of these works generate simulation environments. Code as Policies writes policies with hierarchical code generation, and VoxPoser extracts affordance and constraint maps via LLM-generated code. Can the authors clarify what is meant by this statement?

---

> ### Author Response · Authors · 2025-11-18
>
> **[Q1] Response to "How well does a single-agent iterative refinement framework perform?"**:
>
> - We implemented single-agent iterative refinement baselines where DRDQ-32B receives error feedback and attempts self-correction up to 3 times (matching our EC's retry limit). Results are in Table 1 (main text) and Table 13 (Appendix A.5.6).
>
>   **Key findings**:  (1) Iterative refinement provides modest gains (+3.4-3.8 points); (2) The strongest baseline (DRDQ-32B + SFT + DPO with 3 correction attempts) achieves 41.3 points, remaining **26.4 points below SMRF+SFT+DPO** (67.7); (3) SMRF+SFT alone achieves 55.7 vs. DRDQ-32B + SFT with 3 correction attempts baseline's 39.7 (+16.0).
>
>   **Why the gap?**: Our Error Corrector trains on physics-specific error tuples (instruction, buggy code, error, fix), enabling domain-aware diagnosis. Single-agent refinement often fixes syntax but **misses physical violations** (e.g., adjusting parameters to eliminate errors without ensuring conservation laws). The Simulation Refiner adds preference-aligned optimization. This validates that specialized agents outperform monolithic self-correction.
>
>
>
> **[Q2] Response to "Inaccurate characterization of related works (SimGen, VoxPoser, Code as Policies)"**:
>
> - We sincerely thank the reviewer for catching this inaccurate statement. We have revised the related work section to accurately reflect what these works actually do:
>
>   **Original:** "Robotics applications like SimGen, VoxPoser, and Code as Policies generate simulation environments for robot task planning."
>
>   **Revised:** "Other robotics applications leverage LLMs for task execution: Code as Policies writes hierarchical robot control policies, VoxPoser composes 3D value maps for motion planning via LLM-generated code, and SimGen generates realistic driving scene images. However, these focus on high-level task planning or scene generation within constrained domains."
>
>   We apologize for the confusion. These works do not generate simulation environments. Instead, they use LLMs for different aspects of robot control and scene generation. Our work differs by generating fundamental physics simulation code that symbolically represents physical laws in executable simulation environments (specifically Genesis), which is distinct from generating control policies, value maps, or scene images.

---

> ### Author Response · Authors · 2025-11-28
> **Follow-up on Paper 15733 Rebuttal - Single-Agent Comparison Results Added**
>
> Dear Reviewer LEZS,
>
> Thank you for your insightful review and constructive feedback. We have carefully addressed your concerns in our rebuttal posted on Nov 18, including:
>
> - Added single-agent iterative refinement experiments (Table 1 & 13) showing SMRF's 26.4-point advantage
> - Corrected the characterization of related works (SimGen, VoxPoser, Code as Policies)
>
> With only a few days remaining in the discussion period, we would greatly appreciate if you could review our responses to see if we have adequately addressed your concerns, particularly regarding the single-agent vs. multi-agent comparison you requested.
>
> Your valuable feedback has already helped improve our work significantly, and we look forward to any additional insights you may have.
>
> Best regards,
> Authors of Paper 15733

---

### Meta-Review · Area_Chair_ynMa · 2026-01-05

**Summary:**

This paper introduces PhysCodeBench, a benchmark for physics-aware symbolic simulation of 3D scenes. In this task, an agent is prompted with a natural language instruction and then must generate code to control a physics engine that implemented the desired scene. Towards solving this task, the author design a pipeline to create the required training and evaluation data. They then propose a multi-agent framework that outperforms existing foundation models on this task.

There are two major concerns that the AC believes were not sufficiently addressed in the rebuttal.

The first is that the proposed method is limited to a single physics engine. This raises the question as to whether or not the agent is learning any transferable physical concepts or is simply learning the peculiarities of the given API. While the authors stated that they believe that transferable concepts are being learned, no concrete details or experiments were provided.

The second is that the evaluation metrics are largely superficial and do not measure for any quantitative properties of the resulting physics simulation. This concern was not adequately addressed in the rebuttal.

While there is promise in this paper (as shown by the high score given by Reviewer TMKC), the AC does not believe it can be recommended for publication at ICLR.

**Reviewer Concerns:**

## Reviewer LEZS

- Baselines do not examine using a single model. I believe this addressed.
- Human preference study sample size is small. This was unaddressed.
- Correctness of references. I believe this is addressed.

## Reviewer xguB

- The benchmark is heavily reliant on the high-level APIs exposed by the Genesis physics engine. While the AC appreciates the author's response along these lines, I don't think this concern has been fully addressed.
	- This concern points at a larger questions: What is the gap for current foundation models between their knowledge of physics vs their knowledge of how Genesis works?
- The evaluation metrics are superficial. I believe this is not adequately addressed. While a VLM would catch serious physical implausibilities, it would miss the subtle ones that the existing metrics would also miss.

## Reviewer ezgr

- The tasks are relatively simply. I believe this is addressed
- Metrics are largely qualitative. This is concern is also shared by Reviewer xguB and I don't believe it has been addressed here either.
- The method is limited to Genesis. I don't believe this has been fully addressed. The authors state the in principle the method could be applied to multiple engines and mention an initial experiment with MuJoCo, but no concrete details are provided.

## Reviewer TMKC

- The pipeline requires significant human oversight. This was not addressed.
- Figure 2 is hard to read. This was not addressed.

**Reviewer Scores:**

I do not anticipate that any reviewers would have adjusted their scores. The only score change I can see is Reviewer TMKC decreasing their score if they saw and agreed with the concerns raised by Reviewers ezgr and xguB.

---

### Decision · Program_Chairs · 2026-01-26

Reject